# FIERY1 promotes microRNA accumulation by suppressing rRNA-derived small interfering RNAs in *Arabidopsis*

Chenjiang You[1,2,3], Wenrong He[3], Runlai Hang [1,3], Cuiju Zhang[1], Xiaofeng Cao [4], Hongwei Guo[5], Xuemei Chen [3], Jie Cui [1*] & Beixin Mo [1*]

Plant microRNAs (miRNAs) associate with ARGONAUTE1 (AGO1) to direct post-transcriptional gene silencing and regulate numerous biological processes. Although AGO1 predominantly binds miRNAs in vivo, it also associates with endogenous small interfering RNAs (siRNAs). It is unclear whether the miRNA/siRNA balance affects miRNA activities. Here we report that *FIERY1* (*FRY1*), which is involved in 5′–3′ RNA degradation, regulates miRNA abundance and function by suppressing the biogenesis of ribosomal RNA-derived siRNAs (risiRNAs). In mutants of *FRY1* and the nuclear 5′–3′ exonuclease genes *XRN2* and *XRN3*, we find that a large number of 21-nt risiRNAs are generated through an endogenous siRNA biogenesis pathway. The production of risiRNAs correlates with pre-rRNA processing defects in these mutants. We also show that these risiRNAs are loaded into AGO1, causing reduced loading of miRNAs. This study reveals a previously unknown link between rRNA processing and miRNA accumulation.

[1] Guangdong Provincial Key Laboratory for Plant Epigenetics, Longhua Institute of Innovative Biotechnology, College of Life Sciences and Oceanography, Shenzhen University, 518060 Shenzhen, Guangdong, China. [2] Key Laboratory of Optoelectronic Devices and Systems of Ministry of Education and Guangdong Province, College of Optoelectronic Engineering, Shenzhen University, 518060 Shenzhen, Guangdong, China. [3] Department of Botany and Plant Sciences, Institute of Integrative Genome Biology, University of California, Riverside, CA 92521, USA. [4] State Key Laboratory of Plant Genomics and National Center for Plant Gene Research, Institute of Genetics and Developmental Biology, Chinese Academy of Sciences, 100101 Beijing, China. [5] Department of Biology, Southern University of Science and Technology of China, 518055 Shenzhen, Guangdong, China. *email: huademunaiyi@163.com; bmo@szu.edu.cn

Small RNAs (sRNAs) serve as sequence determinants in post-transcriptional gene silencing (PTGS) in plants. The two major types of PTGS small RNAs are microRNAs (miRNAs) and small interfering RNAs (siRNAs)[1]. Like miRNAs, PTGS siRNAs are usually 21–22 nt long, but unlike miRNAs, they are derived from double-stranded precursors from transgenes, viruses, and endogenous loci, such as the *TRANS-ACTING SIRNA* (*TAS*) loci[1]. The machinery underlying the biogenesis and function of miRNAs and siRNAs contains shared and distinct components[2,3]. miRNA precursors are processed by DICER-LIKE1 (DCL1) into mature miRNAs[4], whereas siRNAs from transgenes, viruses, and endogenous transcripts are generated by other DCLs[2,3,5,6]. Both miRNAs and siRNAs undergo 3′ terminal methylation by HUA ENHANCER1 (HEN1)[7,8]. In addition, both types of sRNAs associate with ARGONAUTE1 (AGO1) to form the functional RNA-induced silencing complex (RISC)[1–3]. The partial sharing of the silencing machinery imply crosstalk and potential mutual regulation between miRNAs and siRNAs.

RNA quality control (RQC) suppresses siRNA production from many endogenous transcripts. Among the RQC genes are those encoding 5′-3′ EXORIBONUCLEASE3 (XRN3) and XRN4; the decapping complex subunits DECAPPING1 (DCP1), DCP2, and VARICOSE (VCS); the SUPERKILLER (SKI) complex components SKI2 and SKI3, which are involved in 3′-5′ exori-bonucleolytic RNA degradation; Nonsense-mediated decay components UPFRAMESHIFT1/3 (UPF1/3); the 3′-5′ POLY(A)-SPECIFIC RIBONUCLEASE (PARN); and exosome subunits RIBOSOMAL RNA PROCESSING4 (RRP4) and RRP6L1[1,9–13]. The sRNAs produced when these RQC genes are compromised are usually 21–22 nt long, phased, and dependent on the PTGS siRNA pathway[10,12]. Zhang et al. proposed that aberrant RNAs accumulated in these mutants are bound by SUPPRESSOR OF GENE SILENCING3 (SGS3)[14] and serve as templates for siRNA biogenesis[15]. Interestingly, a study on the decapping complex demonstrated that DCP1, DCP2, and VCS are required for the accumulation of some miRNAs[16].

Ribosomal RNA (rRNA)-derived sRNAs have been observed in several organisms. In *Schizosaccharomyces pombe*, defects in TRAMP-mediated RNA surveillance trigger the biogenesis of Ago1-associated, rRNA-derived siRNAs (rr-siRNAs)[17]. In *Neurospora crassa*, 20–21 nt qiRNAs are produced from aberrant rRNAs in an RdRP (RNA-dependent RNA polymerase)-dependent manner, are enriched in 5′ U, and are loaded into the AGO protein QDE-2[18]. qiRNAs are thought to function in DNA damage repair. In *Caenorhabditis elegans*, 22-nt rRNA-derived siRNAs (risiRNAs) corresponding to both strands of rDNA are generated under conditions that induce pre-rRNA processing defects. *C. elegans* risiRNAs are enriched in 5′ G and are thought to regulate rRNA abundance[19,20]. In *Arabidopsis*, rDNA-derived 24-nt siRNAs were first described in 2006[21]. These siRNAs are produced in a Pol IV-dependent and DCL3-dependent manner and guide DNA methylation[21,22]. Later studies also found 21-nt small RNAs originating from bidirectional transcripts from the intergenic spacers (IGS) of rRNA genes[23,24]. A recent report found that viral infections trigger the production of siRNAs from rRNAs, but the molecular or biological impacts of these ribosomal small RNAs remain unknown[25].

In this study, a mutation in *FIERY1* (*FRY1*) is isolated in a genetic screen aimed at uncovering new factors in the miRNA pathway. *FRY1* encodes a dephosphorylating enzyme that converts 3′-phosphoadenosine 5′-phosphosulfate (PAPS) into adenosyl 5′-phosphosulfate (APS) in sulfur assimilation[26,27]. FRY1 also converts 3′-phosphoadenosine 5′-phosphate (PAP) into 5′ AMP and Pi. *FRY1* functions in various biological processes, such as stress signaling[27], drought tolerance[28], cell elongation, flowering time[29], leaf development[30], root development[31], and plant

immunity[32]. *fry1* mutant phenotypes resemble those of higher-order *xrn* mutants, possibly because accumulated PAP suppresses XRN enzymatic activity, thereby compromising 5′-3′ RNA degradation[31,33,34]. Several studies have shown that FRY1 and XRNs function in RNA degradation and suppress PTGS in *Arabidopsis*[9,33,35,36]. In a *fry1* mutant, 21-nt sRNAs from the 5′ external transcribed spacer (ETS) of rRNAs accumulate in a DCL2/4-dependent manner[37]. In addition, miRNA processing intermediates accumulate in *fry1*, possibly due to compromised XRN activity[9,38].

We find that *fry1* mutations lead to the accumulation of 21–22-nt siRNAs from mRNAs and rRNAs, transcripts that do not normally undergo siRNA biogenesis. The production of risiRNAs in *fry1*, as well as *xrn2 xrn3* mutants correlate with pre-rRNA processing defects in these mutants. We show that the siRNAs depend on the PTGS siRNA pathway for biogenesis and are loaded into AGO proteins, AGO1 and AGO2. More importantly, risiRNAs compete with miRNAs for these AGO proteins, resulting in the compromised abundance of miRNAs. Removal of risiRNAs partially rescues both the miRNA abundance defects and the plant phenotypes of *fry1*. Collectively, the findings provide insights into the biogenesis of endogenous siRNAs and the crosstalk between siRNAs and miRNAs.

## Results

**A mutation in *FIERY1* was isolated from a suppressor screen.** *CTR1* is a negative regulator in the ethylene response pathway[39]. *ctr1* mutants exhibit constitutive ethylene responses, resulting in shorter root and hypocotyl, tightened apical hook, and swollen hypocotyl[40]. We took advantage of this conspicuous "triple" response phenotype to construct a visual reporter of miRNA activity. We designed an artificial miRNA targeting *CTR1* (amiR-CTR1) driven by a β-estradiol-inducible promoter (Fig. 1a). Upon induction, the transgenic line harboring amiR-CTR1 exhibited the *ctr1* mutant phenotype (Supplementary Fig. 1a), consistent with amiR-CTR1 accumulation and reduced *CTR1* expression (Supplementary Fig. 1b, c). This reporter line was mutagenized by EMS, and M2 plants were screened for mutants resembling wild-type (WT) plants after induction. One mutant, T5520 (Fig. 1b), with a compromised triple response was isolated. As expected, analysis of amiR-CTR1 and CTR1 protein levels in T5520 revealed reduced accumulation of amiR-CTR1 (Fig. 1c) and partial recovery of CTR1 protein levels (Supplementary Fig. 1d) compared to the amiR-CTR1 parental line after β-estradiol induction. T5520 also had pleiotropic phenotypes, including round leaves and delayed flowering (Fig. 1d and Supplementary Fig. 1e). The leaf phenotypes differed from those of canonical miRNA biogenesis mutants such as *ago1* and *hyl1*[41].

To identify the causal mutation in T5520, we conducted whole-genome re-sequencing using pooled plants with the mutant phenotypes in the F2 population of the T5520 x Col-0 cross. A G- > A mutation in AT5G63980 (*FIERY1/SAL1*, *FRY1* hereafter) was identified and verified by genotyping. The mutation was in the acceptor site of the second intron and resulted in the skipping of exon 3 and the creation of an early stop codon (Supplementary Fig. 1f–h). Thus, T5520 is likely a null mutant of *FRY1*. The *FRY1* coding sequence rescued the developmental abnormalities of T5520 (Supplementary Fig. 1i).

**FRY1 promotes the accumulation of miRNAs.** The reduced accumulation of amiR-CTR1 in T5520 suggested that the *fry1* mutation impacted miRNA biogenesis. Unfortunately, the amiR-CTR1 transgene was silenced in T3 and later generations, which prevented further studies of amiR-CTR1. To determine whether FRY1 promotes miRNA accumulation, we examined endogenous

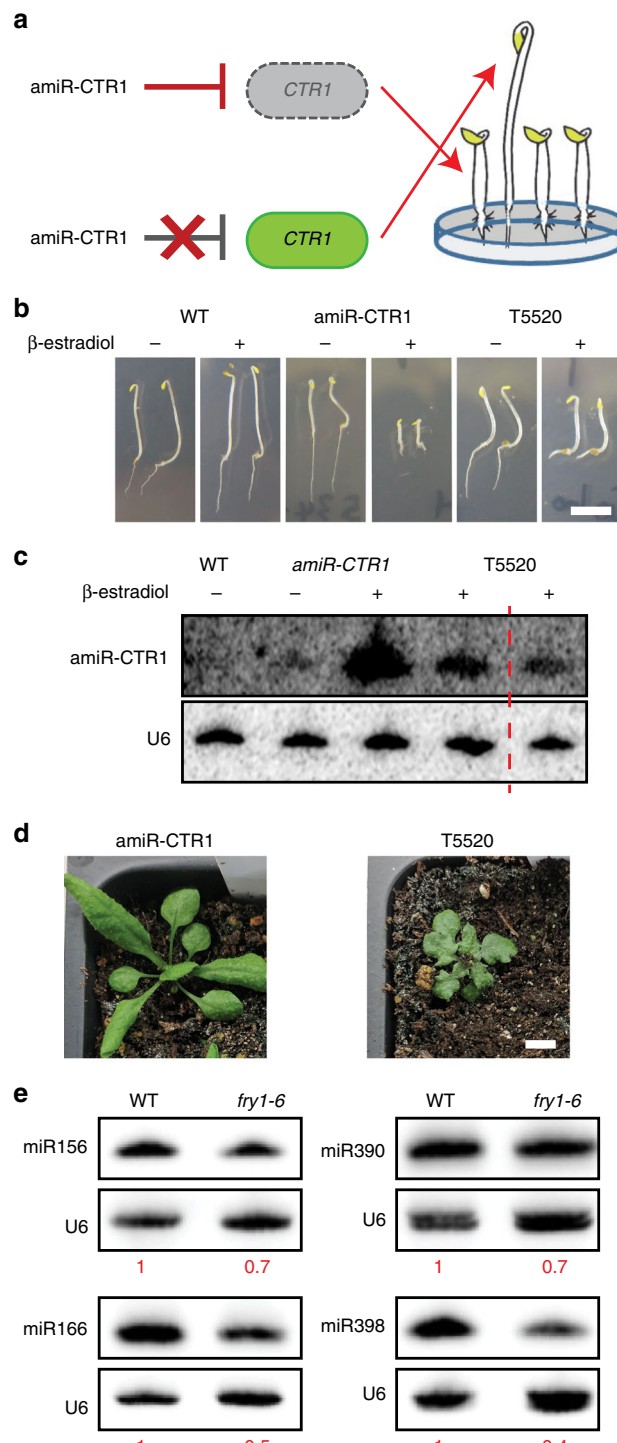

**Fig. 1** FRY1 promotes miRNA accumulation. **a** A diagram of the amiR-CTR1 reporter system. Mutations that disrupt miRNA biogenesis and/or function are expected to impair the regulation of *CTR1* by amiR-CTR1 and, consequently, the "triple response" phenotype. **b** Phenotypes of 5-day-old *Arabidopsis* seedlings. Upon induction, amiR-CTR1 plants exhibited the *ctr1* phenotype, whereas T5520 seedlings failed to show the *ctr1* phenotype. Scale bar = 5 mm. **c** Detection of amiR-CTR1 in T5520. Upon β-estradiol induction, amiR-CTR1 strongly accumulated in amiR-CTR1 plants. However, the accumulation was compromised in T5520 (two biological replicates separated by the red dashed line). **d** Phenotype of T5520 without β-estradiol induction. T5520 plants were smaller than WT and had abnormal leaves. Scale bar = 5 mm. **e** RNA gel blot assay for endogenous miRNAs. All four miRNAs showed reduced accumulation in *fry1-6*. The U6 snRNA was used to determine the relative miRNA levels (as indicated by the numbers below the blots) between the two genotypes. Source data are provided as a Source Data file

miRNAs, 27 and 29 miRNAs were reduced in *fry1-6* and *fry1-8*, respectively, which outnumbered upregulated miRNAs (Fig. 2b). The sRNA-seq data for the abundance changes of miR156, miR166, miR390, and miR398 were similar to those detected by RNA gel blot analysis (Figs. 2c and 1e). miR168 was among 8 upregulated miRNAs identified by sRNA-seq, and RNA gel blot analysis for miR168 showed a similar change in abundance (Fig. 2d). Other upregulated miRNAs included two miR395 family members targeting *APS* genes involved in sulfur metabolism, and the upregulation of miR395 probably might have occurred in response to the altered sulfonation pathway in *fry1*[29,42]. A previous study also reported reduced miRNA levels in *fry1* mutants, but the cause of the defect was not thoroughly investigated[9].

To pinpoint the miRNA biogenesis defects in *fry1* mutants, we first examined the levels of endogenous pri/pre-miRNAs and the expression of major miRNA biogenesis factors by RT-qPCR and protein gel blot assays (Supplementary Fig. 2c–e). For six miRNAs showing reduced levels in *fry1-6* and miR168, which increased in abundance in *fry1-6*, we quantified the corresponding pri/pre-miRNA levels. Only one (pri/pre-miR156a) had reduced accumulation, one (pri/pre-miR390b) had increased accumulation, and pri/pre-miR159b, pri/pre-miR166a, pri/pre-miR167a, pri/pre-miR168 and pri/pre-miR393b were unaffected in *fry1-6* (Supplementary Fig. 2c). Thus, the changes in pri/pre-miRNA levels did not correlate with the abundance of the mature miRNAs, indicating that the global reduction in miRNA abundance in *fry1-6* could not be explained by defects in *MIR* gene transcription. No substantial downregulation of the miRNA biogenesis genes was observed (Supplementary Fig. 2d). Consistent with the increased levels of miR168 and the co-regulation of miR168 and AGO1[43], AGO1 protein levels were increased in both *fry1* mutants (Supplementary Fig. 2e), suggesting a compensation mechanism for reduced accumulation of miRNAs. Taken together, these results suggest that the global reduction in miRNA levels in the *fry1* mutants was not due to a general defect in *MIR* gene transcription or pri/pre-miRNA processing.

**FRY1 prevents the production of ectopic siRNA**. To further evaluate the changes of sRNAs in *fry1* at a global level, all sRNA reads in the three genotypes were mapped to the genome, and their length distribution was examined. As expected, the WT distribution was characterized by a smaller 21-nt peak and a larger 24-nt peak. In both *fry1* mutants, however, the 21-nt peak was enhanced with a concomitant reduction in the 24-nt peak (Fig. 3a), indicating an unexpected increase in 21-nt endogenous sRNAs.

miRNAs in two *FRY1* T-DNA insertion lines, SALK_020882 (previously named *fry1-6*[9]) and SALK_151367 (designated as *fry1-8* hereafter).

RNA gel blot analysis showed that miR156, miR166, miR390, and miR398 had reduced abundance in *fry1-6* (Fig. 1e). To assess the global effects of *fry1* mutations on endogenous miRNAs, we performed sRNA sequencing of WT, *fry1-6*, and *fry1-8* (Supplementary Table 1 and Supplementary Fig. 2a). A general trend of miRNA downregulation was observed in both mutants (Fig. 2a, Supplementary Fig. 2b, and Supplementary Data 1), consistent with the initial finding of reduced amiR-CTR1 levels in T5520. Among the statistically significant differentially expressed

As the global reduction in miRNA accumulation in the *fry1* mutants could not explain the increase in the 21-nt sRNA peak, we investigated changes in other categories of sRNAs. First, we examined the composition of 21-nt sRNAs. In WT, miRNAs constituted the largest category of 21-nt sRNAs in terms of abundance, with rRNA fragments being the second most abundant, followed by sRNAs from genes, *TAS* loci, and transposable elements (TEs). In the *fry1* mutants, miRNA abundance decreased, while the abundance of sRNAs from coding genes and rRNAs increased (Fig. 3b). Notably, rRNA fragments constituted the most abundant category of sRNAs in the mutants. To identify the sources of the differentially accumulating sRNAs, we compared sRNA abundance in 100-bp bins across the genome between *fry1* and WT. Bins with higher and lower sRNA accumulation in *fry1* were referred to as hyper and hypo DSRs (differential sRNA regions), respectively. We found that 21-nt hyper DSRs greatly outnumbered hyper DSRs of other lengths and hypo DSRs of all lengths in both mutants (Fig. 3c). Many miRNA loci were among the 21-nt hypo DSRs (Supplementary Data 2), and the large number of 21-nt hyper DSRs was consistent with the observed increase in total 21-nt sRNAs. Genomic classification of these 21-nt hyper DSRs revealed that most of them corresponded to rRNA regions and non-miRNA genic regions, consistent with the changes in 21-nt sRNA composition (Fig. 3d).

**Aberrant sRNA accumulation from coding genes**. Many 21-nt hyper DSRs overlapped with coding genes (Fig. 3d). To investigate the changes in genic sRNAs, we used annotated genes as units and identified genes with differential sRNA accumulation, which we referred to as DSGs, between WT and the *fry1* mutants. Hyper DSGs, i.e., genes with higher levels of sRNAs in the *fry1* mutants, constituted the vast majority of DSGs (Fig. 3e). The significant overlap in these hyper DSGs between *fry1-6* and *fry1-8* (Fig. 3f and Supplementary Data 3, super exact test *P* value = 0) indicated that *FRY1* suppresses sRNA production from these genes. There were 228 21-nt hyper DSGs in both mutants combined, and although sRNAs derived from these DSGs constituted only ~4% of the total 21-nt sRNAs (Fig. 3g, Supplementary Fig. 3a), they represented over 10% of the total rRNA-depleted 21-nt sRNAs (Supplementary Fig. 3b). It should be noted that most sRNA analyses in the literature ignore rRNA-derived sRNAs, which were included in this study. We selected two hyper DSGs with highly abundant sRNAs to perform RNA gel blot validation of the sRNA-seq results, and the accumulation of 21-nt sRNAs from these two genes was indeed higher in *fry1-6* relative to WT (Fig. 3h).

We next investigated the possible mechanisms of sRNA accumulation from these coding genes. The accumulation of genic sRNAs was previously reported in RNA decay-deficient mutants such as *ein5-1 ski2-3*, *dcp2-1*, *vcs*, and *xrn3-8*[12,16,33]. We re-analyzed the published sRNA-seq data for these mutants using our own pipeline for comparison to *fry1* (*fry1-6* and *fry1-8* combined). First, we examined the 21-nt hyper DSRs in these mutants and confirmed the accumulation of aberrant 21-nt sRNAs from coding genes reported in the original studies (Supplementary Fig. 3c, d). Next, we compared 21-nt hyper DSGs between *fry1* and these mutants (Fig. 3i). The overlap between *fry1* and all of the analyzed mutants was statistically significant, except for the overlap between *fry1* and *xrn3-8* (Supplementary Data 4). As XRN3 is a nuclear exonuclease while the other proteins are thought to act in cytoplasmic RNA decay, the results indicated that the aberrant genic sRNA accumulation in the *fry1* mutants likely occurred in the cytoplasm. It was previously reported that in the *ein5-1 ski2-3*

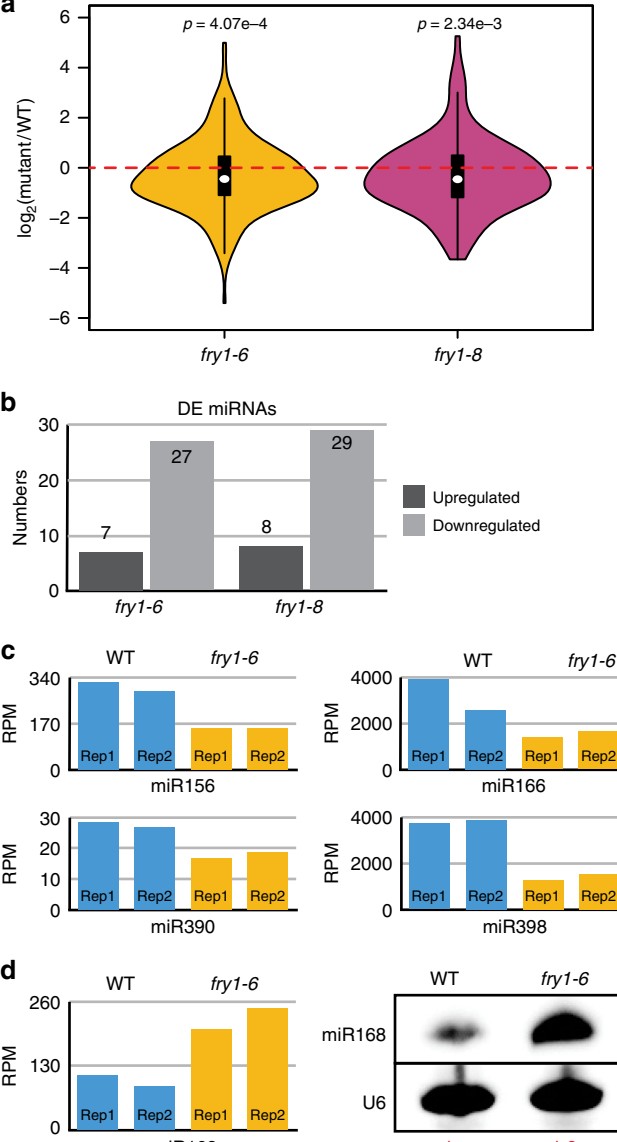

**Fig. 2** Endogenous miRNAs are reduced in *fry1* mutants. **a** Distribution of the fold changes for all detected miRNAs in *fry1-6* and *fry1-8* compared to WT. The black lines in each comparison represent the canonical "box" in a boxplot with the white dot showing the median value. The "violin" shape corresponds to the density of data. *P* values were calculated by a one-sample two-tailed Student's *t* test. The *t* values are −3.5885 and −3.0781, respectively, and the df values are 227 for both comparisons. **b** Numbers of significantly differentially expressed (DE) miRNAs in *fry1* mutants compared to WT. There are a large number of downregulated miRNAs in both *fry1* alleles compared to WT. **c** Normalized read counts of miRNAs in Fig. 1e. Sequencing results (biological replicates are shown separately) of all four downregulated miRNAs, miR156, miR166, miR390, and miR398, are consistent with those from RNA gel blot assays. **d** RNA gel blot validation of upregulated miR168. The left panel shows the normalized read counts from sRNA-seq. The right panel shows the RNA gel blot for miR168 in *fry1-6* compared to WT. The internal control U6 snRNA was used to determine the relative levels of miR168 in the gel blot assay. Source data are provided as a Source Data file

double mutant, 21-nt sRNAs were generated from the 3′ fragments of miRNA target transcripts resulting from miRNA-guided cleavage, due to insufficient exoribonucleolytic degradation of these fragments in the cytoplasm[12]. We examined

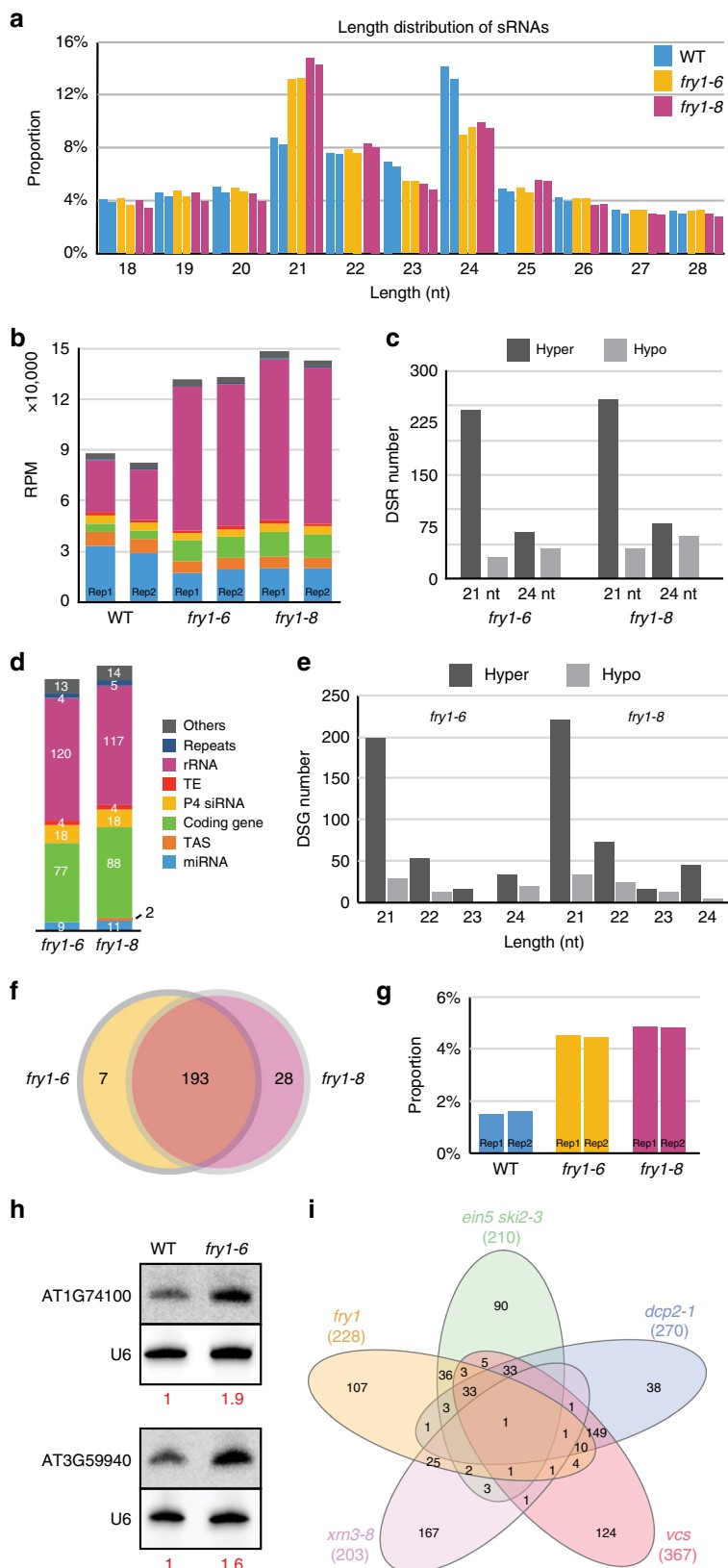

several of these experimentally validated miRNA targets with aberrant 21-nt sRNA accumulation in *ein5-1 ski2-3*, but we found no accumulation of 21-nt sRNAs from these genes in *fry1* (Supplementary Fig. 3e). In contrast, *NIA1* and *NIA2*, which are not targeted by miRNAs, generated aberrant sRNAs in *fry1* (Supplementary Fig. 3e). Moreover, only 17 of the 228 hyper

DSGs in *fry1* are predicted miRNA targets (Supplementary Data 5). We tried to identify common features of the genes that produced 21-nt sRNAs in *fry1* and found that they tended to have fewer exons (Supplementary Fig. 3f) and longer gene length, transcript length, and UTR length (Supplementary Fig. 3g–j).

**Fig. 3** Accumulation of 21-nt sRNAs from coding gene loci in *fry1*. **a** Length distribution of mapped sRNA-seq reads from WT, *fry1-6*, and *fry1-8* seedlings. The 24-nt peak in WT nearly disappeared in both *fry1* mutants. The *Y* axis indicates the percentage of reads of different lengths among the total mapped sRNA reads (18–42 nt). **b** Genomic classification of 21-nt sRNAs in WT and both *fry1* mutants. See **d** for legends. Read counts for miRNAs decreased, and those for sRNAs from rRNAs increased dramatically in *fry1-6* and *fry1-8*. The annotation was adopted from known genome features. The *Y* axis shows the cumulative RPM values for sRNAs corresponding to different features. **c** Number of DSRs in *fry1* mutants compared to WT. Only 21-nt and 24-nt data are shown. The 21-nt hyper DSRs greatly outnumber the other DSRs in both *fry1* mutants. **d** Genomic classification of 21-nt hyper DSRs in both *fry1* mutants. Most 21-nt hyper DSRs in *fry1* corresponded to rRNA and coding genes. **e** Number of DSGs in *fry1* mutants compared to WT. The 21-nt hyper DSGs greatly outnumber other DSGs in both *fry1* mutants. **f** Venn diagram for genes with rogue 21-nt sRNAs in *fry1-6* and *fry1-8*. The overlap between the two sets is 193, which is significant based on a super exact test (*P* value = 0). **g** Proportion of 21-nt sRNAs from 21-nt hyper DSGs among all mapped sRNA reads. The *Y* axis indicates the proportion of 21-nt sRNAs from the combined 21-nt hyper DSGs in both mutants (228 DSGs) among the total mapped sRNA reads. **h** RNA gel blot validation of rogue 21-nt sRNAs from coding genes. Two genes with abundant 21-nt siRNAs, AT1G74100 and AT3G59940, were selected. Though there are bands in WT, the signals increase in *fry1-6*. **i** Venn diagrams for genes with rogue 21-nt sRNAs in *fry1* and previously reported mutants. Except for *xrn3-8*, there was a significant overlap in genes between *fry1* and the analyzed mutants. See Supplementary Data 3 for details. Source data are provided as a Source Data file

**Aberrant sRNA accumulation from rRNAs**. Besides the large number of 21-nt hyper DSRs from coding genes, over half of the 21-nt hyper DSRs were from rRNA regions. rRNA-derived sRNAs are usually considered degradation remnants of rRNAs and have typically been ignored in previous studies of sRNAs in plants. To confirm that the accumulated sRNAs arose from rRNAs and not from other overlapping features, we used the genome browser IGV[44] to visualize the detailed changes in 21-nt sRNAs at rDNA loci in all genotypes. At an rDNA locus on chromosome 3, it was obvious that regions with abundant sRNAs expanded from mature rRNA regions found in WT to the ETS/ITS (external/internal transcribed spacer) regions in *fry1* (Fig. 4a). Moreover, sRNAs were largely from the sense strand in WT, but antisense sRNAs were present in *fry1* (Fig. 4a). To verify the sRNA-seq data, we designed two probes to detect the antisense sRNAs by RNA gel blot analysis, which confirmed the accumulation of these antisense sRNAs from the rDNA locus (Fig. 4b). This finding suggested that the sRNAs were unlikely to be rRNA degradation products. We also wondered whether rRNA-derived sRNAs similarly accumulated in the aforementioned RNA decay mutants. However, in *ein5-1 ski2-3*, *dcp2-1*, *vcs*, and *xrn3-8*, no over-accumulation of sRNAs from rDNA loci (Supplementary Figs. 3d and 4a), particularly from the ETS/ITS regions and the antisense strand (Supplementary Fig. 4b), was observed. Although the sRNA library construction for *ein5-1 ski2-3* and *xrn3-8* included an rRNA removal step, rRNA fragments were still detectable due to incomplete removal. In addition, antisense rRNA fragments were not expected to have been removed by the filtering steps. Thus, the lack of ETS/ITS-derived and antisense strand-derived sRNAs suggested that the mutations in these RNA decay genes did not lead to the production of rRNA-derived sRNAs.

Sequence features, such as length and 5′ nucleotide identity, are highly related to Dicer processing[5] and AGO sorting[45]. We therefore examined these two features of the sRNAs that mapped to rDNA loci to determine whether they are siRNAs. As expected, sRNAs of every length (18–30 nt) arising from the sense strand were nearly equally abundant, suggesting that they correspond to rRNA degradation products. However, only 21-nt sRNAs from both strands dramatically accumulated in *fry1* (Supplementary Fig. 4c), consistent with the finding that the "rRNA" feature was enriched in 21-nt hyper DSRs but not in other size classes (Fig. 3b, c). These findings suggested that these sRNAs might be produced by DCL1 or DCL4, which generate 21-nt sRNAs[5]. In terms of the 5′ nucleotide preference among the 21-nt sRNAs in the mutants, U was the most common 5′ nucleotide among sense sRNAs, while C was the preferred 5′ nucleotide among antisense sRNAs (Supplementary Fig. 4d). The 5′ nucleotide preferences therefore suggested that the aberrantly

accumulated sRNAs were siRNAs that loaded into AGO proteins.

**Defects in 5′-3′ rRNA processing lead to sRNA accumulation**. rRNA-derived sRNAs were observed when Arabidopsis plants were infected with viruses[25]. The production of these siRNAs depends on *RDR1*, which is induced by viral infection[25]. We sought to determine the source and biogenesis requirements of rRNA-derived sRNAs in *fry1* mutants. XRN2, XRN3, and FRY1 are known to be involved in rRNA processing; in *xrn2*, *xrn2 xrn3*, and *fry1* mutants, various forms of aberrant rRNAs accumulate[37,46]. We therefore investigated the integrity of the rRNA processing pathway in the *fry1* mutants and the relationship between aberrant rRNAs and the biogenesis of 21-nt sRNAs. For the analysis, we included the rRNA processing mutants *xrn2-1* and *atprmt3-2* as positive controls[46,47] and *xrn3-2*, which exhibits normal rRNA processing, as the negative control[37,46]. In all of the genotypes, the abundance of mature 25S and 18S rRNAs was similar (Supplementary Fig. 5a). Next, we used well-established probes to examine the ITS regions by RNA gel blot analysis. Consistent with previous reports, both *xrn2-1* and *atprmt3-2* showed changes in rRNA intermediates containing ITS sequences, while *xrn3-2* did not have obvious differences compared to WT (Supplementary Fig. 5b). Interestingly, both *fry1-6* and *fry1-8* had greater accumulation of 35S, 27S, and pre-5.8S rRNAs compared to *xrn2-1* (Supplementary Fig. 5b). These differences may be the consequence of defects in both XRN2 and XRN3 function in *fry1*, as the accumulation of aberrant rRNAs in the *xrn2 xrn3* double mutant was similar to that of *fry1* mutants[37].

We also analyzed the accumulation of miRNAs and rRNA-derived sRNAs in the *xrn2*, *xrn3*, and *xrn2 xrn3* mutants. rRNA-derived sRNAs only accumulated in *fry1-6* and *xrn2 xrn3* (Fig. 4c). In addition, the abundance of miR166 and miR398 was only reduced in *fry1-6* and *xrn2 xrn3* (Fig. 4d), suggesting a negative correlation between the levels of miRNAs and rRNA-derived sRNAs. Intriguingly, changes in the abundance of miRNAs and rRNA-derived sRNAs correlated with the phenotypic severity of the mutants (Supplementary Fig. 5c). However, miR168 showed similar changes in *fry1-6* and *xrn4* and was not affected by mutations in *XRN2* or *XRN3*, suggesting that low XRN4 activity caused the increase of miR168 in *fry1* (Fig. 4e).

A recent study demonstrated that defects in 3′-5′ rRNA processing induced the accumulation of antisense siRNAs from rRNA loci in *C. elegans*[19]. We examined the accumulation of rRNA-derived sRNAs in the *Arabidopsis* exosome mutant *mtr4*, which exhibits defects in 3′-5′ rRNA processing and has marginal effects on PTGS[37,48]. Using the same probes, however, we could

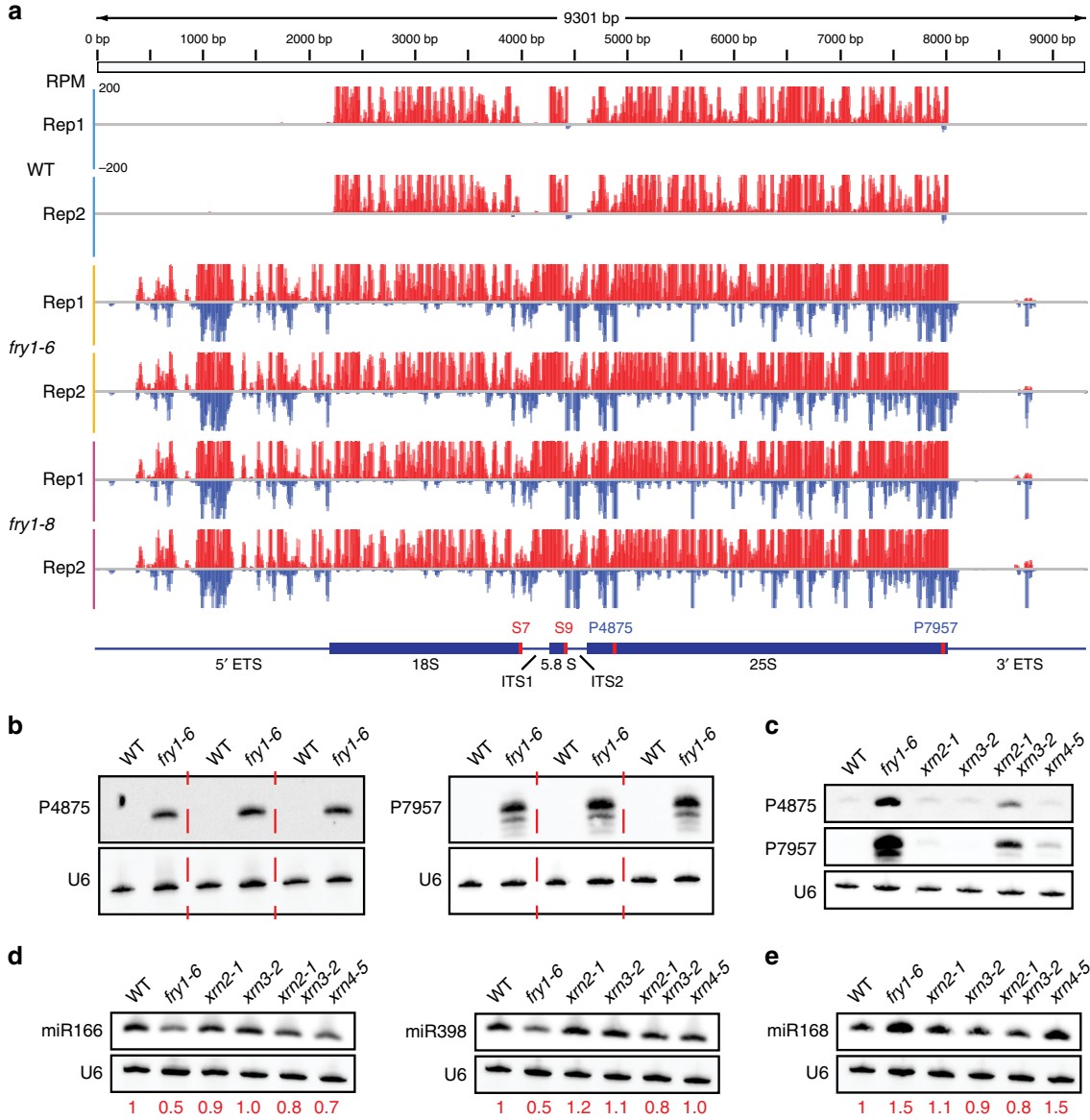

**Fig. 4** 21-nt sRNAs derived from rDNA in *fry1*. **a** An IGV view of 21-nt sRNAs mapped to an rDNA locus on *Arabidopsis* chromosome 3. The red and blue bars represent the normalized read counts from the positive and negative strand, respectively; all *Y* axis ranges in the diagram are −200 to 200. The bottom diagram shows the rDNA locus with the features indicated by black text. The red and blue text indicates the target sites of probes used in this study from the positive and negative strands, respectively. **b** RNA gel blot validation of antisense 21-nt sRNAs derived from rRNAs. Three biological replicates consistently show the accumulation of siRNAs corresponding to the rDNA locus in *fry1-6*. The target sites of probes P6387 and P9469 are shown in **a**. **c** rRNA-derived sRNAs in *xrn* mutants. The rRNA-derived sRNAs were only detected in the *xrn2 xrn3* mutant, but the signal intensity was still lower than that in *fry1-6*. This finding suggests that the rRNA-derived sRNAs are dependent on XRN2/3. **d, e** Abundance of endogenous miRNAs in *xrn* mutants. miR166 and miR398 (**d**) were reduced in *fry1-6* and to a lesser extent in *xrn2 xrn3*. miR168 (**e**) increased in *fry1* and *xrn4* but was not affected by either the *xrn2* or *xrn3* mutation. Source data are provided as a Source Data file

not detect any sRNAs from the analyzed rDNA locus in the *mtr4* mutant (Supplementary Fig. 5d). These findings indicated that the biogenesis of 21-nt sRNAs from aberrant rRNAs in *Arabidopsis* resulted specifically from defects in 5′-3′ exonuclease activity.

**Rogue 21-nt sRNAs are products of the PTGS pathway.** To further investigate whether the coding-gene-derived and rRNA-derived sRNAs were siRNAs, we examined whether their biogenesis required RDR and/or DCL proteins. We crossed *fry1-6* with known siRNA biogenesis mutants, including *rdr1-1*, *rdr2-1*, *rdr6-11*, *dcl2-1*, *dcl3-1*, and *dcl4-2*[49–51]. Intriguingly, the biogenesis of coding-gene-

derived sRNAs and rRNA-derived sRNAs was different. The accumulation of rRNA-derived antisense sRNAs in *fry1-6* was completely suppressed by *rdr6-11*, weakly affected by *rdr2-1* and unaffected by *rdr1-1* (Fig. 5a). Meanwhile, although 21-nt rRNA-derived sRNAs were eliminated in *fry1-6 dcl4-2*, sRNAs of predominantly 22 nt accumulated from the same loci (Fig. 5a). This was due to DCL2, as in the *fry1-6 dcl2-1 dcl4-2* triple mutant, 21–22-nt sRNAs were almost gone (Fig. 5a). However, for coding-gene-derived sRNAs, RDR1 was crucial for their biogenesis as *rdr1-1* suppressed the enhanced sRNA accumulation in *fry1-6* (Fig. 5b). Moreover, these sRNAs accumulated even more in the *fry1-6 rdr6-11* mutant than in *fry1-6*, which might be the consequence of the slightly upregulated *RDR1* expression in the absence of *RDR6*

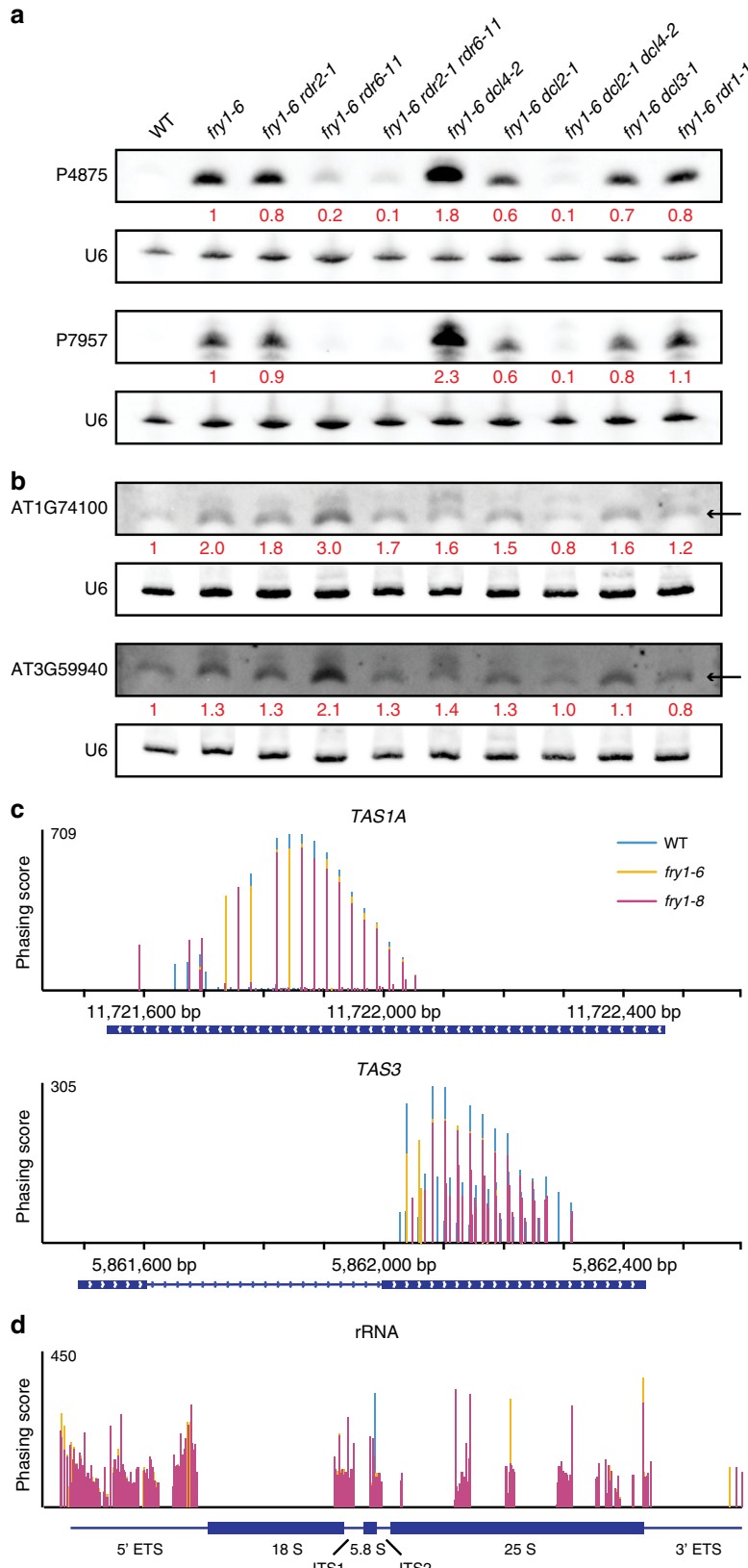

function (Supplementary Fig. 6a). Similar to rRNA-derived sRNAs, *dcl2-1* and *dcl4-2* together completely suppressed the over accumulation of coding-gene-derived sRNAs (Fig. 5b).

As the biogenesis of these 21-nt rogue sRNAs required *RDR1/6* and *DCL2/4*, we concluded that these sRNAs were siRNAs and,

more specifically, ribosomal siRNAs (risiRNAs) for the rRNA-derived ones. The biogenesis requirements of these rogue siRNAs are similar to those of another class of endogenous siRNAs, namely phasiRNAs, which exhibit a head-to-tail phasing signature[3]. Hence, we next investigated the phasing of rogue

**Fig. 5** Rogue 21-nt sRNAs are dependent on *RDR*s and *DCL*s for biogenesis. **a** Accumulation of rRNA-derived sRNAs in *rdr* and *dcl* mutants as determined by RNA gel blot assays. For the antisense sRNAs corresponding to the rDNA locus, *rdr6-11* has a suppressive effect, while *rdr2-1* has only a minor suppressive effect. *dcl4-2* enhances the accumulation of longer sRNAs from these two loci, while in *fry1-6 dcl2-1 dcl4-2*, no accumulation of 21-nt or 22-nt siRNAs was detected. This indicated the antagonistic roles of *DCL2* and *DCL4* in siRNA biogenesis at these loci. **b** RNA gel blot assays to determine the abundance of coding-gene-derived sRNAs in *rdr* and *dcl* mutants. Unlike rRNA-derived sRNAs, these sRNAs largely depend on *RDR1* for biogenesis. Besides, both *DCL2* and *DCL4* are required for the accumulation of coding-gene-derived sRNAs. Black arrows indicate the 21-nt sRNAs. **c**, **d** Regions generating phased siRNAs in WT and *fry1*. At *TAS* genes *TAS1A* and *TAS3* (**c**), phasing scores were slightly reduced in *fry1*. However, there are many phased regions from rDNA detected only in *fry1* (**d**). Source data are provided as a Source Data file

siRNAs using established methods[52]. Surprisingly, the phasing scores at *TAS* and genes known to generate phasiRNAs were slightly reduced in the two *fry1* mutants, while those at the rDNA locus and many 21-nt hyper DSGs were drastically increased (Fig. 5b–d, Supplementary Fig. 6b). These findings reinforced the conclusion that rogue siRNAs, including risiRNAs, were produced from double-stranded RNAs by processive DCL activity.

**21-nt risiRNAs are loaded into AGO1.** To address whether the 21-nt risiRNAs had any biological impacts in vivo, we first examined whether these siRNAs are loaded into AGO1 by immunoprecipitation (IP) of AGO1 in WT and *fry1-6* (Supplementary Fig. 7a), followed by sequencing of the sRNAs from the IP products (Supplementary Data 6). Three independent experiments were performed. As expected, 21-nt sRNAs were enriched and 24-nt sRNAs were depleted in the AGO1 IP sRNA-seq (Fig. 6a). In addition, 5′ U was a predominant feature of sRNAs in both IP products (Supplementary Fig. 7b). This indicated that AGO1 IP sRNA-seq was successful.

As a preliminary analysis, we used the genome browser to examine AGO1 binding of risiRNAs. At the rDNA locus on chromosome 3, 21-nt sRNAs were depleted in AGO1 IP from WT but enriched on both strands, including the ETS/ITS regions, in AGO1 IP from *fry1-6* (Supplementary Fig. 7c). We then conducted IP enrichment analysis in WT and *fry1-6*: IP-enriched bins were defined as 100-bp bins with a statistically significant increase in sRNA abundance in IP versus input. In WT, most enriched bins were from miRNA and other genes, including *TAS* genes, and there were only 7 enriched bins from rRNA regions (Fig. 6b). Compared to WT, the numbers of enriched bins from miRNA and *TAS* genes were similar, but the number of enriched bins from rRNA regions increased in *fry1-6* (Fig. 6b). We also examined the 5′ nucleotides of these risiRNAs (Fig. 6c). In WT, 5′-U sRNAs constituted about 40% of sense and over 80% of antisense 21-nt sRNAs associated with AGO1. In *fry1-6*, the 5′-U percentages were over 90 and 80% for sense and antisense 21-nt sRNAs, respectively. These findings suggested that in WT, many of these sense sRNAs might not be loaded into AGO1 despite their association with it, whereas rogue rRNA-derived sRNAs from both strands in *fry1-6* could be loaded.

Notably, there were also more 21-nt sRNAs and more enriched 21-nt bins from coding genes in *fry1-6* (Fig. 6b). Therefore, we conducted a similar IP enrichment analysis for genes and identified 193 and 1224 genes as those enriched for AGO1-associated 21-nt sRNAs in WT and *fry1-6*, respectively (Fig. 6d). As expected, the 1224 genes in *fry1-6* included most of the 193 genes identified in WT. The 1224 genes also included most of the 228 hyper DSGs identified in *fry1*. Interestingly, the low overlap between the 228 hyper DSGs (i.e., genes with rogue 21-nt sRNAs in *fry1-6* and *fry1-8*) and the 193 genes with AGO1-bound 21-nt sRNAs in WT suggested that the genes suppressed by *FRY1* for sRNA production were distinct from those that generated sRNAs in WT. We also analyzed the sequence features of the 1224 IP-enriched genes in *fry1-6*, and the results supported the hypothesis

that longer genes with fewer exons tended to generate 21-nt siRNAs in *fry1* (Supplementary Fig. 7d–h).

**The loading of miRNAs into AGO1 is compromised in *fry1-6*.** Because most miRNAs associate with AGO1 under normal conditions, the miRNA-binding capacity of AGO1 might be compromised by the excessive accumulation of risiRNAs and siRNAs from other coding genes in *fry1*. To support the hypothesis that rogue siRNAs compete with miRNAs for loading into AGO1, we conducted small RNA-seq following AGO1 IP in *fry1-6 rdr6-11*, in which risiRNAs were barely detectable (Fig. 5a) (Supplementary Table 1 and Supplementary Data 6). As expected, around 50% of the 21-nt sRNAs associated with AGO1 were derived from rRNA in *fry1-6*, while the corresponding proportion in WT was less than 5%, according to the genomic classification of AGO1-associated 21-nt sRNAs (Fig. 7a). Consistent with results from the RNA gel blot assay (Fig. 5a), the proportion of risiRNAs in AGO1 was substantially reduced in *fry1-6 rdr6-11* (Fig. 7a and Supplementary Fig. 8a). In addition, the fraction of AGO1-associated miRNAs decreased to less than 30% in *fry1-6*, and was restored to over 50% by the *rdr6-11* mutation (Fig. 7a). Although the numbers of enriched miRNAs in AGO1 IP were similar in WT, *fry1-6* (Fig. 6b), and *fry1-6 rdr6-11* (Supplementary Fig. 8b), the loading efficiency of miRNAs, as represented by the ratio of miRNA levels in IP and input, was slightly decreased in *fry1-6* and restored in *fry1-6 rdr6-11* (Supplementary Fig. 8c, d and Supplementary Data 7). Specifically, for the 20 most abundant miRNAs in WT, the IP/input ratios were significantly reduced in *fry1-6* (Fig. 7b and Supplementary Fig. 8e, paired Wilcoxon test *P* value = 0.001718) and recovered in *fry1-6 rdr6-11* (paired Wilcoxon test *P* value = 0.003654 between *fry1-6 rdr6-11* and *fry1-6*). These results indicated that in *fry1-6* AGO1′s binding to miRNAs declined, and AGO1 associated with risiRNAs. The reduced miRNA abundance in *fry1* mutants was likely due to the compromised loading of miRNAs into AGO1.

To further support the above hypothesis, we examined the abundance of miRNAs in mutants whose risiRNAs were suppressed or partially suppressed, including *fry1-6 rdr2-1*, *fry1-6 rdr6-11*, and *fry1-6 dcl4-2*. In all these mutants, the down-regulation of miR166 and miR398 was partially suppressed (Fig. 7c). However, the upregulation of miR168 was not affected in any of these double mutants compared to *fry1-6* (Fig. 7d). Furthermore, the *rdr6-11* mutation partially restored the *fry1* mutant phenotype, especially the leaf shape phenotype (Fig. 7e). However, *dcl4-2* did not rescue the *fry1* phenotype, and the *fry1-6 dcl4-2* double mutants were extremely small and died at about 22 days after germination (DAG) (Fig. 7e). This was similar to the phenotype of the *ein5-1 ski2-3 dcl4* triple mutant, in which enhanced biogenesis of 22-nt siRNAs from endogenous genes by DCL2 led to further production of secondary siRNAs[12]. However, while *dcl2 dcl4* can fully rescue the *ein5-1 ski2-3* mutant phenotype[12], the phenotype of the *fry1-6 dcl2-1 dcl4-2* triple mutant was similar to that of *fry1-6* but not WT (Fig. 7e). This also suggested that the developmental phenotypes of *fry1* were not fully attributable to rogue siRNAs.

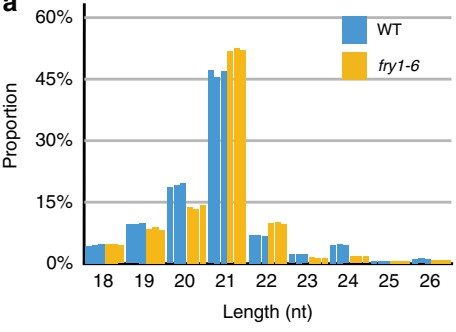

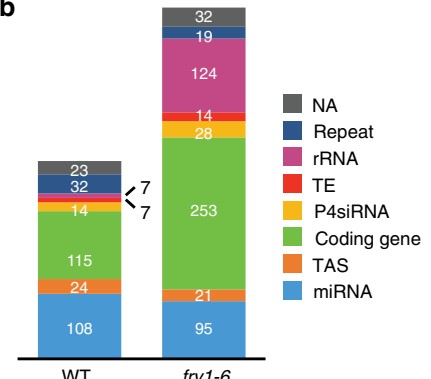

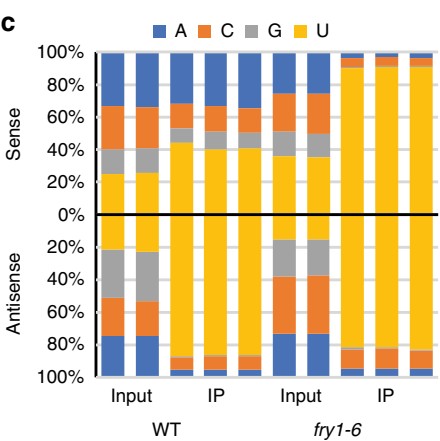

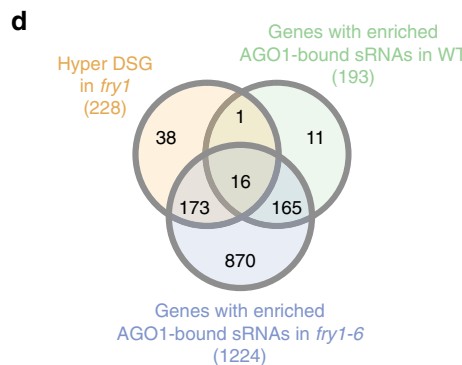

**Fig. 6** Association of 21-nt risiRNAs with AGO1. **a** Length distribution of mapped reads from sRNA-seq of AGO1 IP in WT and *fry1-6*. The proportion of 21-nt reads in all mapped reads is slightly higher in *fry1-6* than in WT. **b** Genomic classification of 100-bp bins with an enrichment of 21-nt sRNAs in AGO1 IP vs. input from WT and *fry1-6*. Although the numbers of bins corresponding to miRNAs and ta-siRNAs are similar between WT and *fry1-6*, those of bins corresponding to coding genes and rRNAs dramatically increased in *fry1-6*. The annotation was adopted from known genome features. **c** Distribution of 5′ nucleotides among 21-nt sRNAs derived from rRNA regions. In WT, a 5′-U preference was only observed among reads from the antisense strand in AGO1 IP, while in *fry1-6*, a 5′-U preference was observed among reads from both strands. This discovery suggests that the majority of sense reads in WT are not bound by AGO1, while those in *fry1-6* can be loaded into AGO1. **d** Venn diagrams for genes with enriched 21-nt sRNAs. In AGO1 IP, most genes with enriched 21-nt sRNAs identified in WT were also identified in *fry1-6*. The 1224 genes in *fry1-6* also included most of the 21-nt hyper DSGs identified in *fry1*. Source data are provided as a Source Data file

*fry1-6*, and this upregulation was reduced in at least one of the *fry1-6 rdr6-11* and *fry1-6 rdr1-1* mutants (Supplementary Fig. 8h). This suggested that mutations in either *RDR6* or *RDR1* can partially restore miRNA activity in the *fry1* mutant. 186 genes accumulating siRNAs in *fry1-6* were detectably expressed by the RNA-seq and they showed different expression patterns in *fry1-6 rdr1-1* and *fry1-6 rdr6-11*. In *fry1-6 rdr1-1*, where siRNA biogenesis from these genes was compromised, the expression of these genes was upregulated significantly as compared to WT. Congruously, their expression decreased in *fry1-6 rdr6-11*, in which siRNA biogenesis from these genes was enhanced (Supplementary Fig. 8i). This indicated that either siRNA biogenesis would eliminate transcripts from these genes or accumulated siRNAs would target the corresponding transcripts for cleavage.

**The loading of miRNAs into AGO2 is compromised in *fry1-6*.** We noticed that miR390, which is predominantly bound by AGO7 and AGO2[53], also decreased in abundance in *fry1-6* (Fig. 1e and Fig. 2c). This may also be attributed to compromised loading of miR390 into AGO2/7 due to competition from rogue siRNAs. Thus, we tested the loading of AGO2-associated miRNAs by AGO2 IP followed by sRNA sequencing in WT, *fry1-6*, and *fry1-6 rdr6-11* (Supplementary Data 6). The composition of 21-nt reads, the most abundant length in AGO2 IP samples (Supplementary Fig. 9a), was very different among the three genotypes. In WT, AGO2 mainly associated with miRNAs and trans-acting siRNAs (ta-siRNAs), while in *fry1-6* risiRNAs and coding-gene-derived siRNAs constituted the majority of AGO2-associated sRNAs (Supplementary Fig. 9b). Meanwhile, as *rdr6-11* removed ta-siRNAs and a portion of risiRNAs, the proportion of AGO2-associated miRNAs increased in *fry1-6 rdr6-11* compared to the *fry1-6* single mutant (Supplementary Fig. 9b).

Next, we conducted analyses to identify genomic regions showing statistically higher levels of sRNAs in AGO2 IP relative to input. In *fry1-6*, the number of enriched bins corresponding to coding genes and Pol IV-dependent siRNA regions drastically decreased while that of rRNA bins increased as compared to WT (Supplementary Fig. 9c). Consistent with the findings from AGO1 IP (Fig. 6b and Supplementary Fig. 8b), the *fry1-6 rdr6-11* mutant also showed a partial restoration of AGO2′s sRNA binding profile in terms of the numbers of enriched bins (Supplementary Fig. 9c). We specifically examined known AGO2-bound miRNAs[53], including miR159, miR390, and miR408 (Supplementary Data 8). The levels of the miRNAs in AGO2 IP decreased in *fry1-6* but were slightly restored in *fry1-6*

Because rogue siRNAs were loaded into AGO1, we sequenced the transcriptome of WT, *fry1-6*, *fry1-6 rdr6-11*, and *fry1-6 rdr1-1* to understand if the siRNAs affected gene expression. Despite the large numbers of upregulated and downregulated genes (Supplementary Fig. 8f, g), we focused on known miRNA targets and the 200 genes generating siRNAs in *fry1-6* (Fig. 3f). Among 10 representatives of known miRNA targets, 7 of them showed upregulated expression in

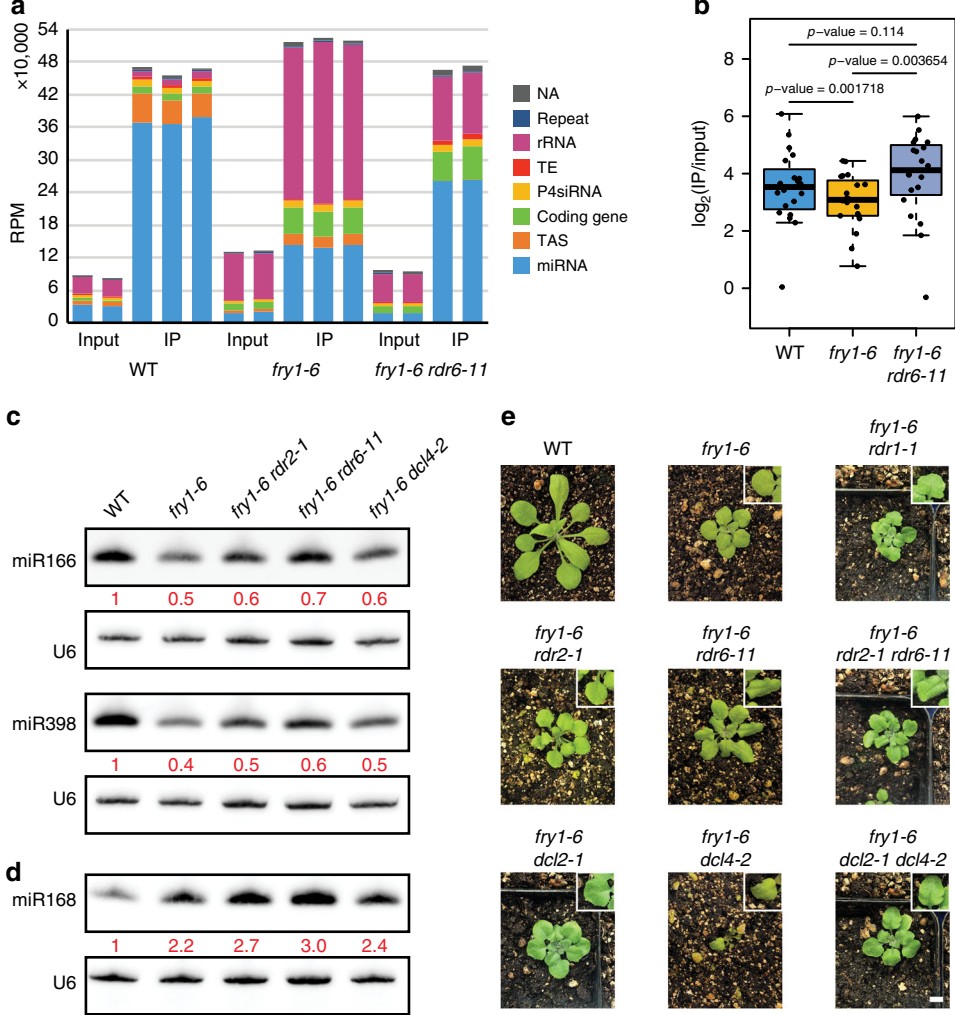

**Fig. 7** *rdr6-11* partially rescues the *fry1* phenotypes. **a** Genomic classification of 21-nt AGO1-associated sRNAs. In WT, miRNAs and ta-siRNAs constitute the majority of AGO1-associated 21-nt sRNAs. In *fry1-6*, there is a drastic increase in rRNA-derived siRNAs, consistent with the total sRNA composition. The *rdr6* mutation results in a partial removal of risiRNAs and a concomitant partial restoration of miRNAs. The *Y* axis shows the cumulative RPM values for sRNAs corresponding to different genomic features. **b** The AGO1 loading efficiency of the 20 most abundant miRNAs in WT. The loading efficiency is represented by the ratio of RPM in immunoprecipitated samples to that in input. The efficiencies in WT and *fry1-6* are significantly different based on a paired Wilcoxon test (*P* value = 0.001718). The efficiencies are recovered by the *rdr6* mutation (*fry1-6 rdr6-11* vs. *fry1-6*: *P* value = 0.003654; *fry1-6 rdr6-11* vs. WT: *P* value = 0.114). **c, d** miRNA accumulation in *rdr* and *dcl* mutants. miR166 and miR398 are downregulated in *fry1-6*, and all three analyzed double mutants show slightly higher abundance of the miRNAs than *fry1-6* (**c**). However, the increased abundance of miR168 in *fry1-6* was still present in the double mutants (**d**). The internal control U6 snRNA was used to determine the relative miRNA levels. **e** Partial rescue of the *fry1* phenotypes by *rdr* and *dcl* mutations. Plants shown are at 22 days after germination. *rdr6-11* can partially rescue the mutant phenotypes of *fry1-6*. Meanwhile, *DCL4* is necessary for the survival of *fry1* mutants, probably due to the enhanced activity of DCL2 in *dcl4-2*. This was supported by the *fry1-6 dcl2-1 dcl4-2* triple mutant. The restoration of leaf shape by *rdr6-11* is shown by the enlarged leaves in the insets. Source data are provided as a Source Data file

*rdr6-11* (Supplementary Fig. 9d). We then examined the coding gene-derived siRNAs and risiRNAs. AGO2 bound siRNAs from 154 of the 228 genes with rogue siRNAs in *fry1* (Supplementary Fig. 9e). Also, 5′ A, which is preferred by AGO2[45], was enriched in risiRNAs from both strands (Supplementary Fig. 9f). We noticed that, although *rdr6-11* suppressed the loading of risiRNAs into AGO2, there were still 80 bins of risiRNAs in *fry1-6 rdr6-11* (Supplementary Fig. 9c, g), and most of them were from the 5′ ETS (Supplementary Fig. 9g). These results suggested that the loading of sRNAs into AGO2 was affected in *fry1-6*.

## Discussion
Because sRNAs from rDNA regions are usually considered degradation products of mature rRNAs, many previous sRNA-

seq analyses have often excluded these sequences by removing them during library construction or after read mapping during data analysis. Still, some studies have reported the existence of rRNA-derived siRNAs and implicated their functions. In *S. pombe*, because rr-siRNAs preferentially begin with 5′ U and associate with Ago1, they may sequester Ago1 and interfere with its function[17]. Recent studies in *C. elegans* have described similar phenomena: under conditions such as 3′-5′ exonuclease impairment, cold stress, and deficient rRNA processing, 22-nt siRNAs with a 5′-G preference accumulated from both strands of rDNA. These risiRNAs can be loaded into NRDE-3, a nuclear Ago protein, and potentially target rRNA precursors[19,20]. In *Arabidopsis*, 24-nt siRNAs from rDNA involved in the RNA-directed DNA methylation pathway was first identified[21,22]. These siRNAs are not derived from rRNA precursors or mature rRNAs as they

are generated in a Pol IV-dependent manner. Small RNAs generated from rRNAs or their precursors have also been found in plants. Some vasiRNAs in virus-infected *Arabidopsis* are from rRNAs, are 21-nt long, and are dependent on DCL4 and RDR1 for biogenesis[25]. Similarly, rRNA-derived 21-nt siRNAs accumulate in *xrn2 xrn3* and *fry1* mutants[37]. But neither of these reports provided evidence for the AGO association of these rRNA-derived small RNAs. In this study, we report the extreme accumulation of AGO1-associated and AGO2-associated risiRNAs in plants. We showed that sRNAs arose from both strands and from the EST/IST regions of the rDNA loci in *fry1* (Fig. 4b) and that they competed with miRNAs for loading into AGO1 and AGO2. Since the accumulation of risiRNAs was dependent on XRN2/3 (Fig. 4c), it is reasonable to assume that their biogenesis occurs in the nucleus. Nuclear risiRNAs would pose a threat to miRNAs, which are loaded into AGO1 in the nucleus[54]. On the other hand, the present findings indicate that risiRNA biogenesis is also dependent on RDR6 and DCL4 (Fig. 5a), which localize to both the cytoplasm and the nucleus[55–57], complicating predictions about the site of risiRNA biogenesis.

Based on the present findings, we propose a competition model as shown in Fig. 8. In WT plants, FRY1 degrades PAP to ensure the activities of XRN4 in the cytoplasm and XRN2/3 in the nucleus. XRN4 and XRN2/3 efficiently degrade aberrant RNAs in the cytoplasm and nucleus, respectively, which prevents the biogenesis of siRNAs. Thus, most AGO1 proteins are occupied by miRNAs. In the *fry1* mutants, PAP accumulates and inhibits the activity of XRNs. Aberrant mRNAs and rRNAs accumulate and are captured by the siRNA pathway, which consequently generates rogue siRNAs that compete with miRNAs to occupy AGO1 and AGO2 proteins. Perhaps as an attempt to reach miRNA homeostasis, AGO1 levels were increased in *fry1* (Supplementary Fig. 2e)[43]. However, excessive risiRNAs and siRNAs from coding genes still outcompeted miRNAs, resulting in their low loading efficiency and reduced abundance. Thus, the proper partitioning of AGO1 for miRNA and siRNA binding requires RQC.

Consistent with previous findings in cytoplasmic RNA decay mutants[12], the *fry1-6 dcl4-2* double mutant was not viable but viability could be restored by the *dcl2-1* mutation (Fig. 7e). Nevertheless, the *fry1-6 dcl4-2 dcl2-1* triple mutant, like *fry1-6 rdr2-1* and *fry1-6 rdr6-11*, still exhibited abnormal phenotypes compared to WT plants (Fig. 7e), indicating only a partial rescue of *fry1* by mutations in the siRNA pathway. Indeed, the abundance of miR166 and miR398 was only partially restored in *fry1-6 rdr6-11* that lacked rogue siRNAs (Fig. 7c). FRY1 and XRN2/3 facilitate the turnover of excised pre-miRNA loops in *Arabidopsis*[9]. This defect in miRNA processing may also affect the abundance of mature miRNAs. The partial rescue of *fry1* by *rdr6-11* or *dcl2-1 dcl4-2* differs from the full phenotypic rescue of *ein5-1 ski2-3* by the same mutations[12], but resembles the minimal rescue of *dcp2-1* and *vcs-6* by *rdr6^{sgs2-110}*. We also noticed that there were still abundant risiRNAs from the 5′ ETS in all *rdr6* samples (Supplementary Fig. S8a and 9g); the biogenesis of these risiRNAs was *RDR6*-independent. These 21-nt risiRNAs may have originated from the bidirectional transcription of rDNA, as previously reported[23] and may account for the partial rescue of *fry1* phenotypes by *rdr6*. Despite this, the *fry1* mutant phenotypes may not be entirely attributable to the accumulation of risiRNAs and 21-nt siRNAs from coding genes. Other processes affected in *fry1*, such as RNA processing, RNA decay, and sulfur metabolism, may also contribute to the phenotypes.

In summary, we show that RNA silencing activity is impacted by competition between miRNAs and siRNAs for AGO1 in *Arabidopsis*, and RNA quality control maintains small RNA homeostasis to ensure proper miRNA activities.

## Methods

**Plant materials and growth**. *Arabidopsis thaliana* wild-type (accession Columbia-0) and T-DNA insertion lines of *FRY1* (AT5G63980, *fry1-6*: SALK_020882, *fry1-8*: SALK_151367), *XRN2* (AT5G42540, *xrn2-1*: SALK_041148), *XRN3* (AT1G75660, *xrn3-2*: SAIL_762H09), *XRN4* (AT1G54490, *xrn4-5*: SAIL_681E01), *PRMT3* (AT3G12270, *atprmt3-2*: WISCDSLOX391A01), *RDR1* (AT1G14790, *rdr1-1*: SAIL_672F11), *RDR2* (AT4G11130, *rdr2-1*: SAIL_1277H08), *RDR6* (AT3G49500, *rdr6-11*: CS24285), *DCL2* (AT3G03300, *dcl2-1*: SALK_064627), *DCL3* (AT3G43920, *dcl3-1*: SALK_005512), and *DCL4* (AT5G20320, *dcl4-2*: GABI_160G05) were used in this study. Seeds were germinated on 1/2 MS medium under short-day conditions (8 h light and 16 h dark) at 22 °C, and seedlings were

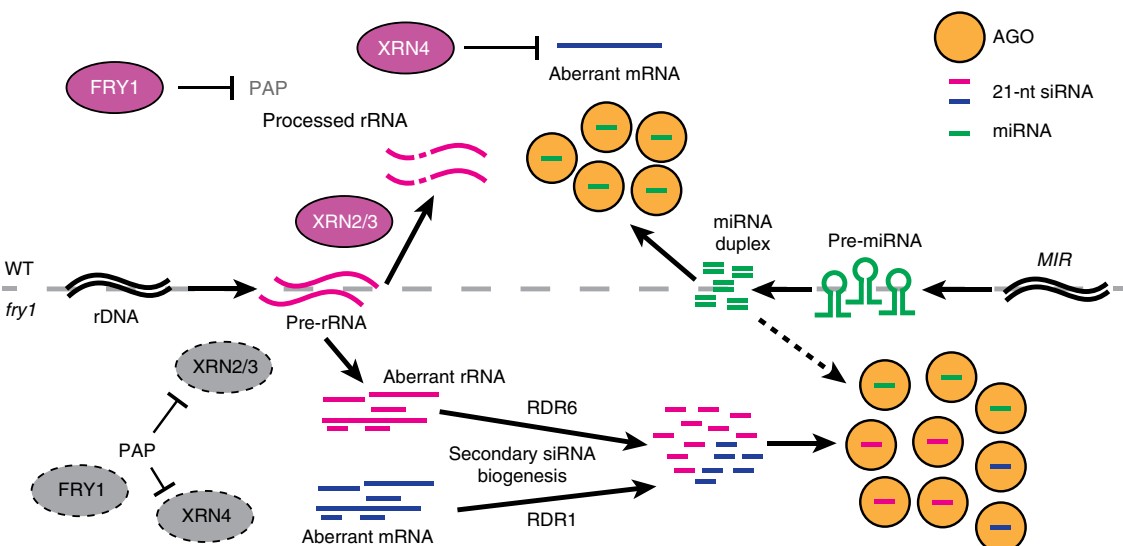

**Fig. 8** A proposed model of *FRY1* function in balancing siRNA and miRNA biogenesis in *Arabidopsis*. In WT plants (upper half of the diagram), FRY1 degrades PAP to ensure the activation of XRN2/3/4 function. XRNs degrade aberrant RNAs to prevent the biogenesis of rogue siRNAs. As a result, most AGO (AGO1 and AGO2) proteins are associated with miRNAs and function in miRNA-directed target regulation. In *fry1* mutants (lower half of the diagram), PAP accumulates and inhibits XRN activity. The resulting aberrant RNAs, including mRNAs and rRNAs, are captured by the PTGS siRNA pathway. Rogue 21-nt sRNAs are generated from the aberrant RNAs by RDR1 and RDR6, respectively, and compete with miRNAs for AGO occupancy. The altered partitioning of AGO between miRNAs and siRNAs leads to reduced abundance of miRNAs

either collected for analyses or transferred to soil for phenotypic observation on day 12.

**EMS mutagenesis and mutant identification**. amiR-CTR1 (CGGGUUGG-GAAUAAUAUGUAU) was designed using WMD tools (http://wmd3.weigelworld.org/cgi-bin/webapp.cgi) then inserted into the *MIR319a* backbone[58]. The amiR-CTR1 fragment was recombined into the plasmid *pER10* containing a β-estradiol-induction cassette using the Xho1 and Spe1 restriction sites (Supplementary Data 9). The construct was transformed into WT *Arabidopsis* by *Agrobacterium* transformation. To locate the T-DNA insertion site, we re-sequenced the amiR-CTR1 transgenic plants on the HiSeq 2000 platform at the genomics core facility at UCR. By mapping the resulting reads to the Araport11 *Arabidopsis thaliana* genome (https://www.araport.org), we determined that the insertion was between nucleotides 18214427 and 18214448 on chromosome 5. We also confirmed that there was only a single T-DNA insertion in the genome, as no chimeric reads from two T-DNA borders (LB/RB) were found. About 2 mL freshly collected amiR-CTR1 seeds were used for ethyl methanesulfonate (EMS) mutagenesis. Seeds were washed in 0.1% Tween-20 for 15 min then treated with 50 mL 0.1% EMS overnight on a rotator in a fume hood. The seeds were transferred to 0.5 M NaOH and incubated overnight, rinsed with ddH$_2$O several times, then washed in ddH$_2$O for 4 h. Finally, the seeds were sown in soil.

To find the causal mutation in T5520, the CTAB method was used to extract DNA from ~50 plants with the T5520 phenotype from the F2 population of the T5520 x Col-0 cross. A DNA library was constructed with the NEBNext Ultra™ II DNA Library Prep Kit for Illumina (E7645S, NEB) according to the manual. The library was sequenced on the HiSeq 2000 platform at the genomics core facility at UCR, and the PE150 reads were mapped to the *Arabidopsis* genome. SNPs were called using SAMtools v1.9[59] then analyzed to identify the causal mutation using an online software NGM (http://bar.utoronto.ca/ngm/index.html). From the NGM result, we narrowed the location of the mutation to a region consisting of 10 candidate genes (AT5G49770, AT5G52170, AT5G54330, AT5G55330, AT5G57060, AT5G62770, AT5G63450, AT5G63980, AT5G64390, and AT5G64430) that contained SNPs with discordant chastity scores over 0.95. We designed dCAPS primers for all of the candidate genes and analyzed another batch of F2 plants with the mutant phenotype[60]. This analysis pinpointed the mutation to AT5G64390 (*FRY1*).

**RNA extraction and northern blotting**. Total RNA was extracted from 12-day-old WT and mutant seedlings using TRI reagent (MRC, TR118) according to the manufacturer's instructions. For each sample, 10 μg total RNA was run on a 15% urea-PAGE gel and transferred to a Hybond NX membrane. The RNA was cross-linked to the membrane with the EDC cross linking buffer (0.16 M EDC, 0.13 M 1-methylimidazole at pH 8.0) at 65 °C for 90 min. Biotin-labeled probes were added to the hybridization buffer (5X SSC, 20 mM Na$_2$HPO$_4$ at pH 7.2, 7% SDS, 2× Denhardt's solution) and incubated with the membrane at 55 °C overnight. After two wash steps (2X SSC, 0.1%SDS, 55 °C, 20 min each time) to remove excess probe, the membrane was processed using the Chemiluminescent Nucleic Acid Detection Module Kit (ThermoFisher, 89880) according to the instruction manual with probes described in Supplementary Data 9. The relative expression level in the RNA gel blots was calculated against the internal control U6 using Fiji[61].

**Protein detection**. Aerial tissues (25 mg) of 12-day-old seedlings were harvested for protein extraction using 1× SDS buffer (100 mM Tris at pH6.8, 4% SDS, 20% Glycerol, 0.2% Bromophenol blue). The samples were loaded onto a 10% SDS-PAGE gel and proteins were then transferred to a nitrocellulose membrane after electrophoresis. Anti-AGO1 antibody (1:3,000, Agrisera, AS09 527) was used to detect the AGO1 protein, and the 60S ribosomal protein L13 was detected with its antibody (1:2,500, Agrisera, AS13 2650) and served as the internal control.

**sRNA library construction and sequencing**. Total RNA (20 μg) from WT (Col-0), *fry1-6*, and *fry1-8* was resolved on a 15% urea-PAGE gel, and the sRNA fraction (15–40 nt) was excised. sRNAs in the excised gel were recovered in 0.4 M NaCl, followed by ethanol precipitation. sRNA libraries were constructed using the NEBNext Small RNA Library Prep Set for Illumina (NEB, E7300S) according to the manufacturer's instructions. The libraries were pooled and sequenced to generate 75-bp single-end reads on an Illumina NextSeq CN500 platform at Berry Genomics (Beijing, China).

**Analysis of sRNA-seq data**. Our sRNA-seq data and published datasets (GSE57936, GSE65056, GSE95473) were analyzed using a publicly available pipeline, pRNASeqTools v0.6. The sRNA-seq raw reads were trimmed to remove the 3′ adapter sequences (AGATCGGAAGAGC) then size-selected (18–42 nt) using cutadapt v1.9[62]. The trimmed reads were mapped to the Araport11 genome using ShortStack v3.4[63] with parameters '-bowtie_m 1000-ranmax 50-mmap u-mismatches 0'. To calculate and compare sRNA abundance in the WT and mutant libraries, the *Arabidopsis* genome was tiled into 100-bp bins (or bins based on specific features, e.g., miRNAs, TEs, genes, and 1000-bp gene-upstream sequences), and reads whose 5′ end nucleotides mapped to a given bin were assigned to that particular bin. Normalization was conducted by calculating the RPM value (reads per million mapped reads) for each bin, and comparison was performed for each

category of bins using the R package DESeq2[64]. The significance of overlap was calculated using the R package SuperExactTest[65]. sRNA target prediction was performed by psRNATarget[66] using the 2017 scheme with expectation ≤2.

**AGO IP and sRNA-seq**. Total protein was extracted from WT, *fry1-6*, and *fry1-6 rdr6-11* seedlings using IP buffer (50 mM Tris 7.5, 150 mM NaCl, 10% glycerol, 0.1% NP-40, and 1× proteinase inhibitor cocktail). AGO1 antibody (8 μL/g) or AGO2 antibody (16 μL/g, Agrisera, AS13 2682) and protein A Dynabeads (Invitrogen, 10002D) were sequentially added to the supernatant to obtain the AGO protein complex. For WT, *fry1-6*, and *fry1-6 rdr6-11*, the immunoprecipitation, RNA extraction, library construction, sequencing, and data analysis were performed in three independent experiments as described above for total sRNAs.

**mRNA-seq and data analysis**. RNA-seq libraries were constructed using NEBNext® Ultra™ RNA Library Prep Kit for Illumina® (NEB, USA) following manufacturer's recommendations, and pooled and sequenced on the Illumina NovaSeq 6000 system (paired-end, 150 bp) at Berry Genomics (Beijing, China). The RNA-seq data were analyzed using the pRNASeqTools. Briefly, raw reads were mapped to the Araport11 genome using STAR v2.6[67] with the parameters "--alignIntronMax 5000 --outSAMmultNmax 1 --outFilterMultimapNmax 50 --outFilterMismatchNoverLmax 0.1". Mapped reads were counted by featureCounts v1.6.4[68] and comparison was performed using the R package DESeq2[64].

**Reporting summary**. Further information on research design is available in the Nature Research Reporting Summary linked to this article.

## Data availability
The raw sequence data generated during this study were deposited into the NCBI GEO database under the accession code GSE133461. The source data for Figs. 1b, c, 2a, c, d, 3a–c, e, g, h, 4b–e, 5a, 6a, c, 7a–e, and Supplementary Figs. 1b–d, 2b–e, 3a–d, f–j, 4a, c, d, 5a–d, 6a, 7a, b, d–h, 8d, e, h, i, and 9a, b, d, f are provided as a Source Data file. The authors declare that any other supporting data is available from the corresponding author(s) upon request.

## Code availability
All bioinformatic analyses in this study were performed by an integrated pipeline for next-generation sequencing analysis, pRNASeqTools v0.6 [https://github.com/grubbybio/pRNASeqTools/]. This pipeline can be used freely under the MIT license.

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

## Acknowledgements

The authors thank Dr. Brandon Le for sharing his ideas in the early stages of this study. This study was supported by the National Natural Science Foundation of China (91740202, 31870287, 31801077) and the Guangdong Innovation Team Project (2014ZT05S078).

## Author contributions

W.H., C.Y., J.C., and X.Chen designed the research. H.G. provided the CTR1 antibody. W.H. performed the screening, and W.H. and C.Y. identified the mutant. W.H. and J.C. prepared the sequencing libraries, and C.Y. conducted the data analyses. J.C., C.Z., R.H., and C.W. performed genetic and biochemical experiments. C.Y., X.Chen, B.M., and X. Cao wrote the paper.

## Competing interests

The authors declare no competing interests.
