## [Peer Review File · Nature Communications]

Reviewers' comments:

Reviewer #1 (Remarks to the Author):

In this manuscript You and colleagues performed a genetic screen using an artificial microRNA (miRNA) construct, isolating a *fry1* mutant allele in Arabidopsis. FIERY (FRY) encodes an enzyme that converts 3'-phosphoadenosine 5'-phosphate (PAP) into 5' AMP and inorganic phosphate in plants. Several past studies have established that *fry1* deficiency mutants phenocopy loss-of-function for XRN family 5'-3' Exoribonucleases in plants (XRN2, XRN3, XRN4). Here, You and colleagues propose that *fry1* mutants promote miRNA accumulation by suppressing rRNA-derived small interfering RNAs (siRNAs).

To investigate this aspect of rRNA-derived siRNA accumulation, the authors used small RNA-seq and northern blot analyses. Past studies found that siRNAs arise from both strands of rDNA repeats and are sometimes, but not always, derived from aberrant rRNAs (Xie et al. 2004 PLoS Biol; Preuss et al. 2008 Mol Cell; Earley et al. 2010 Genes Dev.; Lange et al. 2011 Plant J). rRNA-derived siRNAs (risiRNAs) have already been studied in *fry1* and *xrn2/3* mutants plants, where they over-accumulate (Lange et al. 2011). Interestingly, pre-rRNA processing is also defective in *fry1* and *xrn2/3* plants (Lange et al. 2011). In their Introduction, You and colleagues mention rRNA-derived siRNAs in the other model organisms *N. crassa* and in *C. elegans*, but they do not accurately cite the above plant research on this topic; this includes omitting precedent work from a key paragraph on this subject in their Introduction (Major concern #1, see below).

The authors' finding that FRY1 is important for regulating miRNA abundance in plants is interesting because this may be related to siRNAs over-accumulating in *fry1* mutants. The authors argue that AGO1 is depleted of miRNAs because AGO1 is overloaded with rRNA-derived siRNAs, which could help unravel regulatory processes that balance siRNA versus miRNA occupancy of AGO1. However, the very slight recovery of miR166 and miR398 levels in the *fry1 rdr6* double mutant compared to *fry1* is likely within the error for miRNA quantification via northern (Figure 6g). *fry1 rdr6* double mutant plants, based on the photos presented, remain small with abnormal leaves, like *fry1* (Figure 6i). Such data do not show that *rdr6* clearing of 21-nt risiRNAs restores miRNA occupancy of AGO1 and/or miRNA function to anything resembling a wild-type condition (Major concern #2).

Much of the remaining work in this study is of high technical quality, but it seems better suited to a more specialized journal because Arabidopsis FRY1 has been characterized in several past studies, certain of which already made the connection between risiRNAs and the plant RNA decay factors FRY1, XRN2, XRN3 and XRN4. This is particularly apparent in Figures 3, 4 and 5, which show only incremental advances compared to Lange et al. 2011 and Cao et al. 2014, reports that the authors do not adequately introduce or discuss in their manuscript (Major concern #3).

Major concerns:

1) Page 3-4: Here the authors either omit or neglect to introduce numerous papers that report rRNA-derived siRNAs or rRNA gene-derived siRNAs (risiRNAs) in plants:

- Pontes et al. (2006) Cell
- Preuss et al. (2008) Mol Cell.
- Earley et al. (2010) Genes Dev.
- Pontvianne et al. (2010) PLoS Genet.
- Lange et al. (2011) Plant J.
- Cao et al. (2014) PNAS

Oddly, Lange et al. (2011) is again omitted below in the Introduction (last two paragraphs, Page 4), where the connection between FRY1, XRNs and PTGS are mentioned. All of these studies should be covered in the authors' Introduction, because they are obvious precedents and background for this subfield of RNA silencing.

2) In order to argue that the rRNA-derived siRNAs interfere with miRNA loading, the authors would need to perform a detailed small RNA-seq analysis of AGO1-bound endogenous siRNAs and miRNAs in their *fry1 rdr6* double mutant compared to *fry1* and WT controls.

3) Page 15, line 412: The authors state that, "In this study, we report the extreme accumulation of risiRNAs in RNA decay mutants for the first time in plants." This statement is not accurate. RNA decay mutants have already been shown to over-accumulate risiRNAs in plants: Lange et al. (2011) found that 21-nt risiRNAs over-accumulate in *fry1* and *xrn2/3* mutants. Furthermore, Cao et al. (2014) showed that, upon viral infection, rDNA-derived 21-nt siRNAs hyper-accumulate in *xrn4* mutant plants. Both these papers and their findings merit serious treatment in the current study, because You and colleagues are often reporting the same type of results, while using more numerous northern probes or high-throughput small RNA-seq.

Minor points/corrections

1) Page 2: The "Introduction" section is not labeled in the manuscript, but presumably begins on line 51.

2) Page 2, line 53: The following is not strictly true: "Like miRNAs, PTGS siRNAs are usually 21-22 nt long, but unlike miRNAs, they are derived from aberrant transcripts from transgenes, viruses, and endogenous loci." Not everything that feeds dsRNA synthesis derives from "aberrant" processes. TAS gene products, for example, are not aberrant because they function as precursors for RDR6-dependent dsRNA production in normal development. Certain 21-nt trans-acting siRNAs that are diced from these dsRNAs play an evolutionarily conserved role in phase change and organ development throughout terrestrial plants.

3) Page 4, line 85: The authors state that, "In Arabidopsis, viral infections trigger the production of siRNAs from rRNAs, but the molecular or biological impacts of the risiRNAs remain unknown." However, they fail to cite the relevant paper here: Cao et al. (2014) PNAS. This reference is briefly treated in the Results (Page 10, line 277), but then it is essentially ignored in the authors' Discussion.

4) Figure 4 shows an arbitrarily chosen reference fragment of 45S rDNA corresponding to Arabidopsis thaliana Chromosome 2 for mapping small RNA-seq reads and generating siRNA profile graphics. However, better documented 45S rRNA reference sequences are available in past publications. For example:

- Earley et al. (2010) Genes Dev.
- Pontvianne et al. (2010) PLoS Genet.
- Chandrasekhara et al. (2016) Genes Dev.
- Rabanal et al. (2017) Genome Biol.

Reviewer #2 (Remarks to the Author):

In this study, Mo, Cui and colleagues report that over-accumulation of rRNA-derived siRNAs in the *fry1* mutants reduces miRNA accumulation. They propose that rRNA-derived siRNAs compete with miRNA for AGO1, thereby providing a new layer of miRNA regulation. The manuscript is well written and should be of interest to audience in the related fields. The following issues need to be addressed.

1) A considerable number of miRNAs were up-regulated in the *fry* mutants (Fig. 2b), suggesting that the effect of *Fry1* on miRNA accumulation might have some specificity and/or *Fry1* uses more than one mechanism to regulate miRNA accumulation. The authors should provide more data or discussions to address this issue.

2) miR390 is predominantly loaded into AGO7 and AGO2. Thus, the reduction of its accumulation

in the *fry1* mutant (Fig. 2C) cannot be explained by increased siRNAs competing for AGO1. The authors should consider other mechanisms. At least, they should test if siRNAs also compete for AGO7 and AGO2.

3) The authors propose that Fry1 degrades PAP to protect the activities of XRNs to suppress the production of siRNAs, thereby maintaining miRNA accumulation. If this is the only mechanism through which Fry1 regulates siRNA biogenesis and miRNA accumulation, one would predict similar increase of siRNA production and reduction of miRNA accumulation in the *xrn2/xrn3* mutants; however, compared to that in *fry1*, much less changes were detected in *xrn2/xrn3* (Fig 4d). Actually, a recent study showed that a few miRNAs were upregulated in the *xrn2* mutant (Fang et al., Dev, Cell, 2019). The authors should consider other mechanisms.

Reviewer #3 (Remarks to the Author):

You et al. started to isolate new factors in the miRNA pathway by a genetic screening from synthetic experimental conditions. The synthetic experiment took advantage of the phenotypic changes with or without visible ethylene triple responses and artificial miRNA was designed to promote ethylene-mediated shorter roots and hypocotyls. If a factor that is involved in miRNA mediated regulation was mutated, the effect of amiRNA was dampened and the plant growth would change. They actually could isolate one mutant, segregated its locus in F2 population showing the phenotype and conducted GWAS analysis to find the causal mutation to AT5G63980, that is FIERY gene. The content is to study and understand why the FIERY mutation led to the phenotypic changes. What happened was that the mutant abortively accumulated ribosomal RNA-derived siRNAs which then occupied certain levels of AGO1-bound sRNAs and excluded certain amount of regular miRNAs which were expected to have biological functions in cells.

The main part of experiments and texts are well-organized and pinpointed some neglected facts that ribosomal RNA-related siRNAs exist in cells and are depleted from normal cells not to disturb regular miRNA-mediated gene regulation mainly in AGO1 complex. The results would attract many researchers in the field from now on and encourage some more studies in this field. I hope that discussing such effects would produce some new viewpoints.

L.123: Any quantitative data for reduced fertility? Figures cited were showing delayed flowering.

L.223: No experimental support for “*fry1* mutants likely occurred in the cytoplasm” ?

L.235: Any speculation for a specific preference of FRY1 to fewer exons/longer gene length/long UTR gene?

L.271: “U was the most common 5’ nucleotide among sense sRNAs, while C was the preferred 5’ nucleotide among antisense sRNAs”..... that loaded into AGO proteins.” Did the statement mean that AGO proteins other than AGO1 are involved as well. If so, what do authors speculate on the effect of non-AGO1 bound sRNAs?

L.331: Does the gel pattern described in the previous paragraph show strong evidence for requirement of RDR1 but not for RDR2 ?

L.360: increased in *fry1-6* (Fig.6b)

L.392: the “reshuffle” describes what the authors thought exactly?

L.448: More description/explanation on sulfur metabolism.

L.537: proteinase *inhibitor* cocktail

Reviewer #1 (Remarks to the Author):

In this manuscript You and colleagues performed a genetic screen using an artificial microRNA (miRNA) construct, isolating a *fry1* mutant allele in *Arabidopsis*. FIERY (FRY) encodes an enzyme that converts 3'-phosphoadenosine 5'-phosphate (PAP) into 5' AMP and inorganic phosphate in plants. Several past studies have established that *fry1* deficiency mutants phenocopy loss-of-function for XRN family 5'-3' Exoribonucleases in plants (XRN2, XRN3, XRN4). Here, You and colleagues propose that *fry1* mutants promote miRNA accumulation by suppressing rRNA-derived small interfering RNAs (siRNAs).

To investigate this aspect of rRNA-derived siRNA accumulation, the authors used small RNA-seq and northern blot analyses. Past studies found that siRNAs arise from both strands of rDNA repeats and are sometimes, but not always, derived from aberrant rRNAs (Xie et al. 2004 PLoS Biol; Preuss et al. 2008 Mol Cell; Earley et al. 2010 Genes Dev.; Lange et al. 2011 Plant J). rRNA-derived siRNAs (risiRNAs) have already been studied in *fry1* and *xrn2/3* mutants plants, where they over-accumulate (Lange et al. 2011). Interestingly, pre-rRNA processing is also defective in *fry1* and *xrn2/3* plants (Lange et al. 2011). In their Introduction, You and colleagues mention rRNA-derived siRNAs in the other model organisms *N. crassa* and in *C. elegans*, but they do not accurately cite the above plant research on this topic; this includes omitting precedent work from a key paragraph on this subject in their Introduction (Major concern #1, see below).

The authors' finding that FRY1 is important for regulating miRNA abundance in plants is interesting because this may be related to siRNAs over-accumulating in *fry1* mutants. The authors argue that AGO1 is depleted of miRNAs because AGO1 is overloaded with rRNA-derived siRNAs, which could help unravel regulatory processes that balance siRNA versus miRNA occupancy of AGO1. However, the very slight recovery of miR166 and miR398 levels in the *fry1 rdr6* double mutant compared to *fry1* is likely within the error for miRNA quantification via northern (Figure 6g). *fry1 rdr6* double mutant plants, based on the photos presented, remain small with abnormal leaves, like *fry1* (Figure 6i). Such data do not show that *rdr6* clearing of 21-nt risiRNAs restores miRNA occupancy of AGO1 and/or miRNA function to anything resembling a wild-type condition (Major concern #2).

Much of the remaining work in this study is of high technical quality, but it seems better suited to a more specialized journal because *Arabidopsis* FRY1 has been characterized in several past studies, certain of which already made the connection between risiRNAs and the plant RNA decay factors FRY1, XRN2, XRN3 and XRN4. This is particularly apparent in Figures 3, 4 and 5, which show only incremental advances compared to Lange et al. 2011 and Cao et al. 2014, reports that the authors do not adequately introduce or discuss in their manuscript (Major concern #3).

Major concerns:

- 1) Page 3-4: Here the authors either omit or neglect to introduce numerous papers that report rRNA-derived siRNAs or rRNA gene-derived siRNAs (risiRNAs) in plants:
- Pontes et al. (2006) Cell

- Preuss et al. (2008) Mol Cell.
- Earley et al. (2010) Genes Dev.
- Pontvianne et al. (2010) PLoS Genet.
- Lange et al. (2011) Plant J.
- Cao et al. (2014) PNAS

Oddly, Lange et al. (2011) is again omitted below in the Introduction (last two paragraphs, Page 4), where the connection between FRY1, XRNs and PTGS are mentioned. All of these studies should be covered in the authors' Introduction, because they are obvious precedents and background for this subfield of RNA silencing.

We did not cite the first four papers reporting 24-nt rDNA-derived siRNAs involved in rDNA methylation. The reason is that the 24-nt siRNAs are generated, and function, differently from the risiRNAs that we report here. However, the reviewer is correct in that we should be more inclusive in Introduction to cover all types of small RNAs from rDNA loci. We have added a paragraph in the "Introduction" section to introduce rDNA-derived 24-nt siRNAs. The other two papers, Lange 2011 and Cao 2014, were cited in the original manuscript in "Results" (originally lines 272-276). During this study, we actually consulted Dr. Shou-Wei Ding for details of vasiRNAs reported in the Cao *et al.* paper. We have added a paragraph discussing these two papers in both "Introduction" (lines 87-90) and "Discussion" (lines 481-484).

2) In order to argue that the rRNA-derived siRNAs interfere with miRNA loading, the authors would need to perform a detailed small RNA-seq analysis of AGO1-bound endogenous siRNAs and miRNAs in their *fry1 rdr6* double mutant compared to *fry1* and WT controls.

We performed small RNA-seq of AGO1 IP in the *fry1 rdr6* double mutant. Corresponding results were added into the second to the last subsection of "Results". Briefly, a large portion of risiRNAs was removed by *rdr6* and consistently, AGO1 loading efficiency of miRNAs was restored in the *fry1 rdr6* double mutant. We also performed RNA-seq in *fry1-6*, *fry1-6 rdr1*, and *fry1-6 rdr6*. Results showing that genes generating siRNAs in *fry1-6* were regulated by genic siRNAs were added into this subsection as well. We also conducted a set of IP-seq of siRNAs bound by AGO2, which showed very similar results. These results were included in the last subsection of "Results". For the phenotypes of plants, we discussed that the phenotypes were not solely contributed by the accumulation of risiRNAs, as *fry1* also affected other aspects of RNA biology, such as rRNA processing. Consistently, the *mir4* phenotype also cannot be rescued by *dcl2 dcl4*, according to the Lange 2011 paper. However, the abnormal leaf shape phenotype of *fry1-6* was indeed rescued by *rdr6*.

3) Page 15, line 412: The authors state that, "In this study, we report the extreme accumulation of risiRNAs in RNA decay mutants for the first time in plants." This statement is not accurate.

RNA decay mutants have already been shown to over-accumulate risiRNAs in plants: Lange et al. (2011) found that 21-nt risiRNAs over-accumulate in *fry1* and *xrn2/3* mutants. Furthermore, Cao et al. (2014) showed that, upon viral infection, rDNA-derived 21-nt siRNAs hyper-accumulate in *xrn4* mutant plants. Both these papers and their findings merit serious treatment in the current study, because You and colleagues are often reporting the same type of results, while using more numerous northern probes or high-throughput small RNA-seq.

Sorry for the priority statement in the previous version. We had considered only small RNAs bound by AGO1 as small interfering RNAs (siRNAs). The previous two papers didn't show the AGO-binding of the 21-nt rRNA-derived small RNAs. Actually, in the Cao 2014 paper, the

authors did not find AGO1 or AGO2 association for those rRNA-derived vasiRNAs. Thus, we didn't consider them as risiRNAs. But this was perhaps too narrow a definition for risiRNAs. We have deleted the priority statement and changed the sentence to "In this study, we report the extreme accumulation of AGO-associated risiRNAs in plants."

Minor points/corrections

1) Page 2: The "Introduction" section is not labeled in the manuscript, but presumably begins on line 51.

Added as suggested (line 51).

2) Page 2, line 53: The following is not strictly true: "Like miRNAs, PTGS siRNAs are usually 21-22 nt long, but unlike miRNAs, they are derived from aberrant transcripts from transgenes, viruses, and endogenous loci." Not everything that feeds dsRNA synthesis derives from "aberrant" processes. TAS gene products, for example, are not aberrant because they function as precursors for RDR6-dependent dsRNA production in normal development. Certain 21-nt trans-acting siRNAs that are diced from these dsRNAs play an evolutionarily conserved role in phase change and organ development throughout terrestrial plants.

We have deleted the word "aberrant" as suggested (line 55).

3) Page 4, line 85: The authors state that, "In Arabidopsis, viral infections trigger the production of siRNAs from rRNAs, but the molecular or biological impacts of the risiRNAs remain unknown." However, they fail to cite the relevant paper here: Cao et al. (2014) PNAS. This reference is briefly treated in the Results (Page 10, line 277), but then it is essentially ignored in the authors' Discussion.

We have added this reference and discussed more about it in "Discussion" (lines 485-487).

4) Figure 4 shows an arbitrarily chosen reference fragment of 45S rDNA corresponding to Arabidopsis thaliana Chromosome 2 for mapping small RNA-seq reads and generating siRNA profile graphics. However, better documented 45S rRNA reference sequences are available in past publications. For example:

- Earley et al. (2010) Genes Dev.
- Pontvianne et al. (2010) PLoS Genet.
- Chandrasekhara et al. (2016) Genes Dev.
- Rabanal et al. (2017) Genome Biol.

As suggested, we used the single copy 45S rDNA on chromosome 3 (14195483-14204860) as our reference, as reported by the Rabanal et al. paper. We have changed all the results related to the rDNA structure and coordinates accordingly.

Reviewer #2 (Remarks to the Author):

In this study, Mo, Cui and colleagues report that over-accumulation of rRNA-derived siRNAs in the fry1 mutants reduces miRNA accumulation. They propose that rRNA-derived siRNAs compete with miRNA for AGO1, thereby providing a new layer of miRNA regulation. The

manuscript is well written and should be of interest to audience in the related fields. The following issues need to be addressed.

1) A considerable number of miRNAs were up-regulated in the *fry* mutants (Fig. 2b), suggesting that the effect of *Fry1* on miRNA accumulation might have some specificity and/or *Fry1* uses more than one mechanism to regulate miRNA accumulation. The authors should provide more data or discussions to address this issue.

For hyper miRNAs (8 total), 3 of them are not 21-nt long (miR167c-3p, miR5642, and miR831-3p), which also reflected the compromised AGO1 loading of 21-nt miRNAs in *fry1*. Two miRNAs, miR845a and miR845b, are pollen specific (Borges, 2018 Nat Gen) and trigger the accumulation of easiRNAs. Previous studies have shown that miR845s are up-regulated in rice and tomato under drought stress (Zhou et al., 2010 J Exp Bot, Candar-Cakir et al., 2016 Plant Biotechnol J). Thus, we propose that miR845s were up-regulated in the *fry1* mutant due to the hyper drought-tolerance of the mutant (Wilson et al., 2009 Plant J). The rest two are miR395 family members that target *APS* genes, which are involved in sulfur metabolism. They were likely up-regulated by a feed-back loop responding to the altered sulfonation pathway in *fry1*. The last one is miR168a-3p, which is increased in abundance in the *fry1* mutant probably due to the increased accumulation of the AGO1 protein.

2) miR390 is predominantly loaded into AGO7 and AGO2. Thus, the reduction of its accumulation in the *fry1* mutant (Fig. 2C) cannot be explained by increased siRNAs competing for AGO1. The authors should consider other mechanisms. At least, they should test if siRNAs also compete for AGO7 and AGO2.

One possibility is that increased risiRNAs compete with miR390 for loading into AGO7 or AGO2. Because there are no good antibodies for AGO7, we chose to test the competition model with AGO2. We performed AGO2 IP-seq and found that the loading of miR390 into AGO2 was compromised in *fry1*, and this can be partially restored by *rdr6*. This result, which indicates that the excess 21-nt risiRNAs in *fry1* might also compete with small RNAs for other AGO proteins, was added as the last subsection of "Results". And the model has also been modified.

3) The authors propose that *Fry1* degrades PAP to protect the activities of XRNs to suppress the production of siRNAs, thereby maintaining miRNA accumulation. If this is the only mechanism through which *Fry1* regulates siRNA biogenesis and miRNA accumulation, one would predict similar increase of siRNA production and reduction of miRNA accumulation in the *xrn2/xrn3* mutants; however, compared to that in *fry1*, much less changes were detected in *xrn2/xrn3* (Fig 4d). Actually, a recent study showed that a few miRNAs were upregulated in the *xrn2* mutant (Fang et al., Dev, Cell, 2019). The authors should consider other mechanisms.

We agree that in *xrn2 xrn3* there should also be a slight reduction in miRNA accumulation. Actually in Fig. 4d, there is a ~20% reduction in the levels of miR166 and miR398. We think this smaller decrease in miRNA accumulation is because risiRNAs accumulated to a lesser degree in *xrn2 xrn3* (Fig. 4c) and because no coding-gene-derived siRNAs accumulated. Also, as the previous study (Gy 2007) showed, the pre-miRNA loops accumulated in *fry1*. The authors proposed that these loops might affect the efficiency of pri- or pre-miRNA processing and thus decrease the levels of mature miRNAs. We think this might be another mechanism through which the levels of mature miRNAs are affected in *fry1* mutants. We have added a paragraph in "Discussion" (lines 515-519).

In Fang 2019, the authors detected increased levels of both pri-miRNAs and mature miRNAs in *xrn2-1*. We examined the levels of pri/pre-miRNAs using RT-qPCR and mature miRNA levels with RNA gel blot assays in several independent experiments. However, we could not fully repeat Fang's results. We could only detect increased precursor levels, including miR397, miR398, and miR408, in *xrn2-1 xrn3-2*. Meanwhile, we confirmed that these miRNAs except for miR408 were not affected at the transcription or processing level in *fry1-6*, and that the levels of the mature miRNAs decreased in *fry1-6* and this decrease was partially restored by *rdr6-11*. These results were consistent with our model and are shown below. We thought the discrepancy might be because these miRNAs are highly stress-related and our growth condition was perhaps different from Fang's.

As we didn't detect the accumulation of risiRNAs in *xrn2-1*, it is very possible that in *xrn2-1*, the dominant effect on miRNAs is transcriptional, as proposed by Fang 2019, while in *fry1* (equal to a weak *xrn2 xrn3 xrn4* triple mutant), AGO1 loading is impaired and thus increased pri-miRNA levels (we also detected some up-regulated pri-miRNAs in Fig. S2) cannot lead to an increase in the levels of mature miRNAs.

Figure legend:

Levels of miRNA precursors and mature miRNAs in various genotypes as indicated. (a-c) Three independent RT-qPCR experiments to determine the levels of miRNA precursors including pri/pre-miR397, pri/pre-miR398, and pri/pre-miR408. Except for pri/pre-miR408, which was strongly up-regulated in *fry1-6*, the precursors of miR397 and miR398 were only up-regulated in *xrn2-1 xrn3-2*. In the third experiment (c), we included more miRNA precursors. However, all of them except for pri/pre-miR408, including pri/pre-miR159b, pri/pre-miR168a, pri/pre-miR390b, pri/pre-miR393b, and pri/pre-miR857, showed either reduced or similar accumulation in all mutants as compared to WT. d RNA gel blot assays to determine the levels of mature miRNAs in various mutants. miR397, miR398, and miR408 all showed reduced levels in *fry1-6* and the reduction was partially restored in *fry1-6 rdr6-11*. However, none of them were up-regulated in *xrn2-1* or *xrn2-1 xrn3-2*.

Reviewer #3 (Remarks to the Author):

You et al. started to isolate new factors in the miRNA pathway by a genetic screening from synthetic experimental conditions. The synthetic experiment took advantage of the phenotypic changes with or without visible ethylene triple responses and artificial miRNA was designed to promote ethylene-mediated shorter roots and hypocotyls. If a factor that is involved in miRNA mediated regulation was mutated, the effect of amiRNA was dampened and the plant growth would change. They actually could isolate one mutant, segregated its locus in F2 population showing the phenotype and conducted GWAS analysis to find the causal mutation to AT5G63980, that is FIERY gene. The content is to study and understand why the FIERY mutation led to the phenotypic changes. What happened was that the mutant abortively accumulated ribosomal RNA-derived siRNAs which then occupied certain levels of AGO1-bound sRNAs and excluded certain amount of regular miRNAs which were expected to have biological functions in cells.

The main part of experiments and texts are well-organized and pinpointed some neglected facts that ribosomal RNA-related siRNAs exist in cells and are depleted from normal cells not to disturb regular miRNA-mediated gene regulation mainly in AGO1 complex. The results would attract many researchers in the field from now on and encourage some more studies in this field. I hope that discussing such effects would produce some new viewpoints.

L.123: Any quantitative data for reduced fertility? Figures cited were showing delayed flowering. The reduced fertility phenotype only exists in T5520 but not in *fry1-6* or *fry1-8*. Thus, this phenotype might be the additive effect of *FRY1*^{G->A} and other mutations in T5520. We have decided to delete this statement to not be misleading.

L.223: No experimental support for “fry1 mutants likely occurred in the cytoplasm”? According to the previous studies on the subcellular localization of XRN2/3/4 and the decapping complex, XRN2 and XRN3 are in the nucleus, and XRN4, DCL1/2, and VCS are in the cytoplasm. Thus, because genes generating ectopic siRNAs in *fry1* were more similar to those in *ein5 ski2*, *dcp2*, and *vcs*, we proposed that the biogenesis of these genic siRNAs in *fry1* was dependent on XRN4 and the decapping complex in the cytoplasm.

L.235: Any speculation for a specific preference of FRY1 to fewer exons/longer gene length/long UTR gene?

According to the speculation of the Zhang 2015 paper, genes generating coding transcript-derived siRNAs (ct-siRNAs) in the *ein5 ski2* mutant are highly expressed genes. And some of the genes that generate siRNAs in *fry1* mutants are also likely expressed at a high level, including genes involved in photosynthesis, and these genes are usually single-exon. The other genes with longer gene length may be called because they have longer transcripts to generate enough siRNAs meeting our threshold.

L.271: “U was the most common 5’ nucleotide among sense sRNAs, while C was the preferred 5’ nucleotide among antisense sRNAs that loaded into AGO proteins.” Did the statement

mean that AGO proteins other than AGO1 are involved as well? If so, what do authors speculate on the effect of non-AGO1 bound siRNAs?

It's possible that other AGO proteins were also affected by excess 21-nt siRNAs. We have performed an IP-seq assay to test the loading of AGO2. And indeed, the siRNAs can also be loaded into AGO2. We have added the last subsection of "Results" to describe this and modified our model.

L.331: Does the gel pattern described in the previous paragraph show strong evidence for requirement of RDR1 but not for RDR2?

We reached the conclusion by following results:

- 1) The level of siRNAs reduced to a WT-like level in the *fry1 rdr1* double mutant.
- 2) The level of siRNAs remained similar to that in WT in the *fry1 rdr2* double mutant.
- 3) The level of siRNAs increased in *fry1 rdr6* mutant where *RDR1* was up-regulated.

L.360: increased in *fry1-6* (Fig.6b)

Added as suggested (line 369).

L.392: the "reshuffle" describes what the authors thought exactly?

With "reshuffle" we would like to suggest a huge change of the composition of small RNA associated with AGO1. We have changed the sentence (lines 408-410).

L.448: More description/explanation on sulfur metabolism.

Added more information about *fry1*'s role in sulfur metabolism as suggested in "Introduction" (lines 94-96).

L.537: proteinase inhibitor cocktail

Added as suggested (line 618).

REVIEWERS' COMMENTS:

Reviewer #1 (Remarks to the Author):

You and colleagues have thoroughly revised their manuscript on how *Arabidopsis thaliana* *fry1* mutants promote miRNA accumulation by suppressing rRNA-derived siRNAs (risiRNAs). I had three major concerns about the original manuscript. My first concern was that key background literature on rRNA-derived siRNAs was not covered. The authors now introduce and discuss this literature more attentively. My second concern was that northern blots and plant leaf phenotypes were not sufficient evidence to argue that miRNA occupancy of AGO1 is restored in *fry1 rdr6* double mutant plants. The authors performed additional sequencing of small RNAs associated with AGO1 in this background, satisfying my concern (Figure 7(a)). Finally, I had initially seen this investigation as offering an incremental advance compared to past work on risiRNAs in plants. The authors' new data, extended bioinformatic analyses, revised discussion and detailed responses to all the referees' comments have fully convinced me of the importance of this study to the broader scientific community.

Reviewer #2 (Remarks to the Author):

The authors have addressed most of my concerns by providing experimental data or explanations.

Reviewer #3 (Remarks to the Author):

It took much efforts and time to look over the revised ms comparing with the original submission. And the reviewer found that revised text and relevant data were not so significantly modified to show the reproducibility/quantification of the data.

I noticed that Fig5d and Fig.7b were exceptionally substituted. Is there any explanation for the facts?

REVIEWERS' COMMENTS: Reviewer #1 (Remarks to the Author): You and colleagues have thoroughly revised their manuscript on how *Arabidopsis thaliana fry1* mutants promote miRNA accumulation by suppressing rRNA-derived siRNAs (risiRNAs). I had three major concerns about the original manuscript. My first concern was that key background literature on rRNA-derived siRNAs was not covered. The authors now introduce and discuss this literature more attentively. My second concern was that northern blots and plant leaf phenotypes were not sufficient evidence to argue that miRNA occupancy of AGO1 is restored in *fry1 rdr6* double mutant plants. The authors performed additional sequencing of small RNAs associated with AGO1 in this background, satisfying my concern (Figure 7(a)). Finally, I had initially seen this investigation as offering an incremental advance compared to past work on risiRNAs in plants. The authors' new data, extended bioinformatic analyses, revised discussion and detailed responses to all the referees' comments have fully convinced me of the importance of this study to the broader scientific community. Reviewer #2 (Remarks to the Author): The authors have addressed most of my concerns by providing experimental data or explanations. Reviewer #3 (Remarks to the Author): It took much efforts and time to look over the revised ms comparing with the original submission. And the reviewer found that revised text and relevant data were not so significantly modified to show the reproducibility/quantification of the data. I noticed that Fig5d and Fig.7b were exceptionally substituted. Is there any explanation for the facts? Because we used a different rDNA reference in this revised manuscript, Fig 5d was replotted. We didn't have a Fig 7b in the original version. The Fig 7b in this version was newly added.